# 3D Scene Assertion Verification

**Jun Lin**[1][*]  **Jiayu Ding**[2][*]  **Xiangtian Si**[1]  **Xitong Cao**[3]
**Lixin Hong**[1]  **Zhang Chen**[1]  **Chenxi Lv**[1]  **Wenqian Wang**[4]

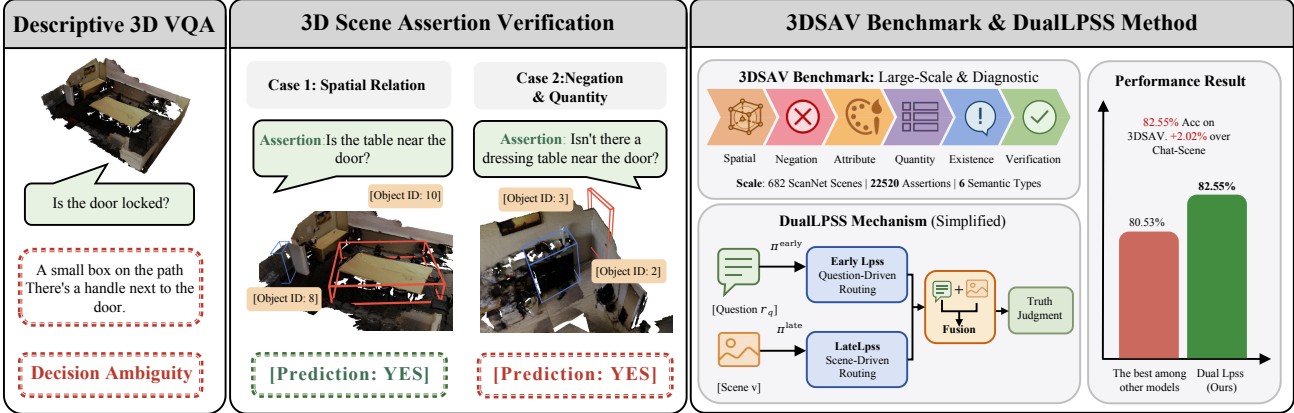

*Figure 1.* **Overview of the proposed task, benchmark, and method.** (Left) We introduce the task of **3D Scene Assertion Verification**. (Middle) We construct **3DSAV**, the first large-scale diagnostic benchmark. (Right) Our **DualLPSS** achieves state-of-the-art accuracy.

## Abstract

Existing 3D Visual Question Answering (3D-VQA) methods rely on generative outputs that can be ambiguous in decision-making settings. We introduce **3D Scene Assertion Verification**, a task that verifies natural language assertions in 3D scenes with strict binary judgments. We present **3DSAV**, a large-scale diagnostic benchmark with 22.5k samples across six semantic types. To address this task, we propose **DualLPSS**, which uses dual-stage subspace routing for type-aware cross-modal fusion and scene-guided assertion focusing. Experiments show that DualLPSS achieves state-of-the-art performance on 3DSAV and handles complex logical assertions better than existing 3D-VQA baselines.

## 1. Introduction

Verifying physical states from hypotheses is a core capability of AI (Lake et al., 2017; Puyin et al., 2025; Wang et al., 2025; Jin et al., 2026b). Safe interaction relies on explicit, actionable judgments rather than ambiguous descriptions (Ahn et al., 2022; Gao et al., 2022; Ren et al., 2023). In 3D perception and Embodied AI, agents must verify the truthfulness of scene properties against natural language (Majumdar et al., 2024). However, traditional descriptive outputs often lack structure, hindering downstream planning. In contrast, planning systems typically model interactions via "Binary Preconditions." For instance, AR systems verify plane emptiness to avoid collisions, while robots confirm target occlusion before grasping. In these scenarios, only binary results serve as valid inputs for logical closed loops, ensuring trustworthy task execution.

However, existing research paradigms do not directly target this need for "judgment." The most relevant task, 3D Visual Question Answering (3D-VQA), centers on generating natural language to provide descriptive answers. This paradigm has practical limitations in verification scenarios requiring explicit decisions. **(i)** It introduces decision ambiguity. Generative models tend to output declarative facts rather than conclusive judgments (Li et al., 2023; Jin et al., 2026a). Given the query "Is the path from the door to the fridge clear?", a 3D-VQA model might reply, "There is a small box on the path." While factually correct, this is invalid information for a robot needing to decide "whether

---

[*]Equal contribution  [1]School of Future Technology, China University of Geosciences, Wuhan, Hubei, China [2]School of Electronic and Computer Engineering, Peking University, Shenzhen, Guangdong, China [3]School of Geography and Information Engineering, China University of Geosciences, Wuhan, Hubei, China [4]Department of Psychology, University of the Chinese Academy of Sciences, Beijing, China. Correspondence to: Jiayu Ding <jyding25@stu.pku.edu.cn>.

*Proceedings of the 43rd International Conference on Machine Learning*, Seoul, South Korea. PMLR 306, 2026. Copyright 2026 by the author(s).

to proceed immediately," as it requires a secondary judgment on whether the "small box" constitutes an obstruction. **(ii)** The generation-based training paradigm is not directly optimized for logical discrimination. Existing 3D-VQA models typically optimize using cross-entropy loss over a full vocabulary (Hong et al., 2023; Huang et al., 2024; Xu et al., 2024). Mathematically, this maximizes the conditional likelihood of the next token, guiding the model to fit the linguistic statistical distribution of training data and reduce perplexity. Consequently, this prioritizes linguistic plausibility over the accuracy of logical judgment. Applying this sequence-probability-oriented paradigm to verification tasks that require convergence to deterministic logical states results in a misalignment between optimization objectives and task requirements (Jin et al., 2025; Li et al., 2023).

In light of these limitations, we advocate shifting the research paradigm from "generative description" to "logical verification." To this end, we formally define the task of **3D Scene Assertion Verification**. Unlike traditional QA, this task returns a deterministic truth value for a supplied statement about a 3D scene, so the output can be used directly by downstream decision logic. To support this setting, we construct the **3D S**cene **A**ssertion **V**erification (**3DSAV**) benchmark. As the first large-scale diagnostic benchmark of its kind, it comprises 22,520 fine-grained annotated samples built upon ScanNet, spanning six semantic types: spatial relationships, negation, attributes, quantities, existence, and verification. Some 3D-QA questions can be answered with yes or no, but 3DSAV asks a different question: given an assertion, is it true in the scene? This is similar to the distinction between QA and natural language inference, where similar surface forms test different reasoning behavior.

However, we observe that existing 3D-VQA models exhibit suboptimal performance in this specific context, and notably, there are no established baselines tailored for assertion verification. To bridge this gap, we propose **Dual-stage Layered Progressive Subspace Specialization** (DualLPSS). A visual overview of the overall task, benchmark, and method is shown in Figure 1. DualLPSS uses dual-stage subspace routing to guide cross-modal fusion and focus the text on scene evidence, so verification is explicit rather than open-ended.

The contributions of this paper are as follows:

- We formally define the task of 3D Scene Assertion Verification.
- We construct the first large-scale benchmark dataset for this task, **3DSAV**, containing 22,520 human-annotated assertions across 682 3D scenes.
- We propose **DualLPSS** for this task. By leveraging dual-stage subspace routing, it enhances logical judgment accuracy.
- Extensive experiments on 3DSAV demonstrate that our method achieves state-of-the-art performance.

## 2. Related Work

**3D Scene Understanding** is a rapidly evolving field aiming to achieve precise comprehension and interaction within 3D environments via natural language. The domain primarily encompasses three core tasks: *3D Visual Grounding* (Achlioptas et al., 2020; Chen et al., 2020; Ding et al., 2025), which localizes objects based on textual queries as seen in datasets like ScanRefer (Chen et al., 2020), Nr3D, and Sr3D (Achlioptas et al., 2020); *3D Dense Captioning*, requiring dense object localization and description, with representative methods including Scan2Cap (Chen et al., 2021), D3Net (Chen et al., 2022a), and Vote2Cap-DETR (Chen et al., 2023); and *3D Visual Question Answering* (Azuma et al., 2022; Ma et al., 2023), typified by systems such as ScanQA and SQA3D. Furthermore, the field spans various other tasks (Yang et al., 2023; Ding et al., 2026; Lv et al., 2024; Li et al., 2026; Zhu et al., 2025b; Pan et al., 2025), such as embodied navigation, scene graph generation, and point cloud segmentation. However, existing paradigms predominantly focus on generative description or localization, lacking verification tasks that yield explicit truth judgments.

**3D Visual Question Answering** generally follows two paradigms: specialized architectures and Large Language Models (LLMs). Regarding specialized architectures, early works prioritized task-specific designs that limited cross-task generalization. Conversely, recent research explores unified frameworks, such as 3D-VisTA (Zhu et al., 2023) which develops a generalist framework via pre-training for indoor 3D QA; BridgeQA (Mo & Liu, 2024) which bridges multi-view RGB features and point clouds to answer scene-level questions; FE-3DGQA (Chen et al., 2022b) which utilizes language-conditioned spatial reasoning for 3D object grounding and QA; and BUTD-DETR (Jain et al., 2022) which formulates 3D Visual Grounding as detector-style object proposal reasoning. In the realm of LLMs, 3D-LLM (Hong et al., 2023) incorporates location embeddings but relies on 2D encoders, often failing to fully capture 3D spatial structures. Chat-Scene (Huang et al., 2024) unifies tasks through object identifiers to achieve state-of-the-art performance, LLaVA-3D (Zhu et al., 2025a) extends multi-view vision-language pretraining to 3D scenes for generative QA, and Video3D-LLM (Zheng et al., 2025) learns position-aware video representations. Nevertheless, these methods still rely on uniform cross-modal fusion and generative outputs. This makes them less suited to decision-making settings that require deterministic verification.

## 3. Task Definition and Benchmark

### 3.1. Task Definition

We formally define the task of *3D Scene Assertion Verification*. Given a 3D scene representation $\mathcal{S}$ (e.g., a reconstructed point cloud) and a natural language assertion $\mathcal{A}$, the

*Table 1.* **Comparison with existing 3D question answering datasets.** Our dataset focuses on logical reasoning verification rather than information retrieval.

| Dataset | Ours | ScanQA | SQA3D |
|---|---|---|---|
| QA pairs | 22,520 | 30,238 | 29,884 |
| Scenes | 682 | 633 | 583 |
| Answer type | Yes/No | Free-form | Free-form |
| *Reasoning complexity (%)* | | | |
| Negation | **19.5** | 0.2 | 0.2 |
| Disjunction | **10.0** | 0.3 | N/A |
| Quantitative | **16.6** | 4.5 | 14.0 |
| Multi-object ($\geq$2) | **68.1** | 29.5 | N/A |
| *Task paradigm* | | | |
| Deterministic output | ✓ | ✗ | ✗ |
| Logical verification | ✓ | ✗ | ✗ |
| Semantic type labels | ✓ | ✗ | ✗ |

objective is to learn a mapping function $\mathcal{F} : (\mathcal{S}, \mathcal{A}) \to y$, where $y \in \{\texttt{Yes}, \texttt{No}\}$ represents the truth value of the assertion relative to the scene.

### 3.2. Dataset

#### 3.2.1. DATASET CONSTRUCTION

The proposed 3DSAV benchmark is built upon ScanNet (Dai et al., 2017). We select 682 scenes for assertion generation, yielding 22,520 samples across **six** semantic types: *spatial* (spatial relationships), *negation* (negative logic and scope), *attribute* (observable visual properties), *quantity* (counting and quantification), *existence* (object presence), and *verification*, which covers comprehensive assertions.

We construct the dataset using a two-stage pipeline. **(1) LLM Generation.** We utilize Qwen3-VL-Plus (Bai et al., 2025) to generate candidate assertions with scene-aware prompts containing two key components: scene context (object lists and target type distribution ratios) to ensure scene relevance, and generation constraints requiring verifiability through observable scene information while excluding external knowledge (see Appendix A and G for prompt templates and 42 quality constraints). **(2) Human Verification.** Eight experts with 3D vision backgrounds conduct verification via a custom 3D visualization platform. The procedure involves truth correction (verifying and rectifying Yes/No labels) and sample cleaning (filtering invalid samples with hallucinations, external knowledge dependence, or semantic ambiguity). Each assertion undergoes independent verification by at least two annotators. Inter-annotator agreement was measured on all reviewed samples before filtering; samples without complete agreement were discarded. Table A1 summarizes the annotation guidelines.

#### 3.2.2. COMPARISON WITH EXISTING BENCHMARKS

As shown in Table 1, 3DSAV differs fundamentally from ScanQA and SQA3D in both task paradigm and reasoning requirements. **(i) Logical Reasoning**: 3DSAV exhibits substantially higher proportions of complex logical structures. *Negation* assertions (19.5%) require parsing the scope of negation operators (e.g., "Is the bookshelf *not* next to any chair?") before verification, while ScanQA contains merely 0.2%. Assertions containing *disjunctions* (10.0%) involve compound conditions with logical OR (e.g., "Are there at least 2 chairs *or* at least 3 tables?"), requiring systematic evaluation of multiple sub-conditions, whereas ScanQA contains only 0.3%. *Quantitative* assertions (16.6%) demand precise numerical verification with operators like "at least", "exactly", and "more than", contrasting with ScanQA's descriptive "how many" questions. Assertions involving *two or more objects* (68.1%) require reasoning over multiple entities simultaneously, compared to ScanQA's 29.5%. **(ii) Task Paradigm**: 3DSAV enforces deterministic binary outputs, eliminating the ambiguity inherent in free-form generation. Furthermore, 3DSAV is designed for logical verification rather than information retrieval, where the model must determine truth values through reasoning rather than merely extracting scene descriptions. Each sample also carries a semantic type label for diagnostic evaluation.

#### 3.2.3. DATASET STATISTICS

3DSAV contains 22,520 assertions across 682 scenes. The label distribution is nearly balanced, with 48.0% Positive (Yes) and 52.0% Negative (No) samples, effectively mitigating potential biases from class imbalance. The overall distribution across semantic types is as follows: spatial (24.7%), negation (19.5%), attribute (18.4%), quantity (16.6%), existence (11.5%), and verification (9.3%). Detailed statistics on logical structures, assertion complexity, and object distributions are provided in Appendix A. For experimentation, we partition the dataset into training (547 scenes, 18,620 pairs), validation (68 scenes, 2,001 pairs), and testing (67 scenes, 1,899 pairs) sets. We ensure strictly disjoint scene splits to prevent data leakage.

### 3.3. Evaluation Metrics

We formulate 3DSAV as a binary classification task requiring deterministic Yes/No predictions. To evaluate overall performance, we report Accuracy, Precision, Recall, and Macro-F1 scores, where Macro-F1 is computed over the binary Yes/No classes. Furthermore, to provide fine-grained diagnostic insights, we report the classification accuracy for each of the six semantic categories: spatial reasoning, negation understanding, attribute recognition, quantity estimation, existence detection, and comprehensive verification.

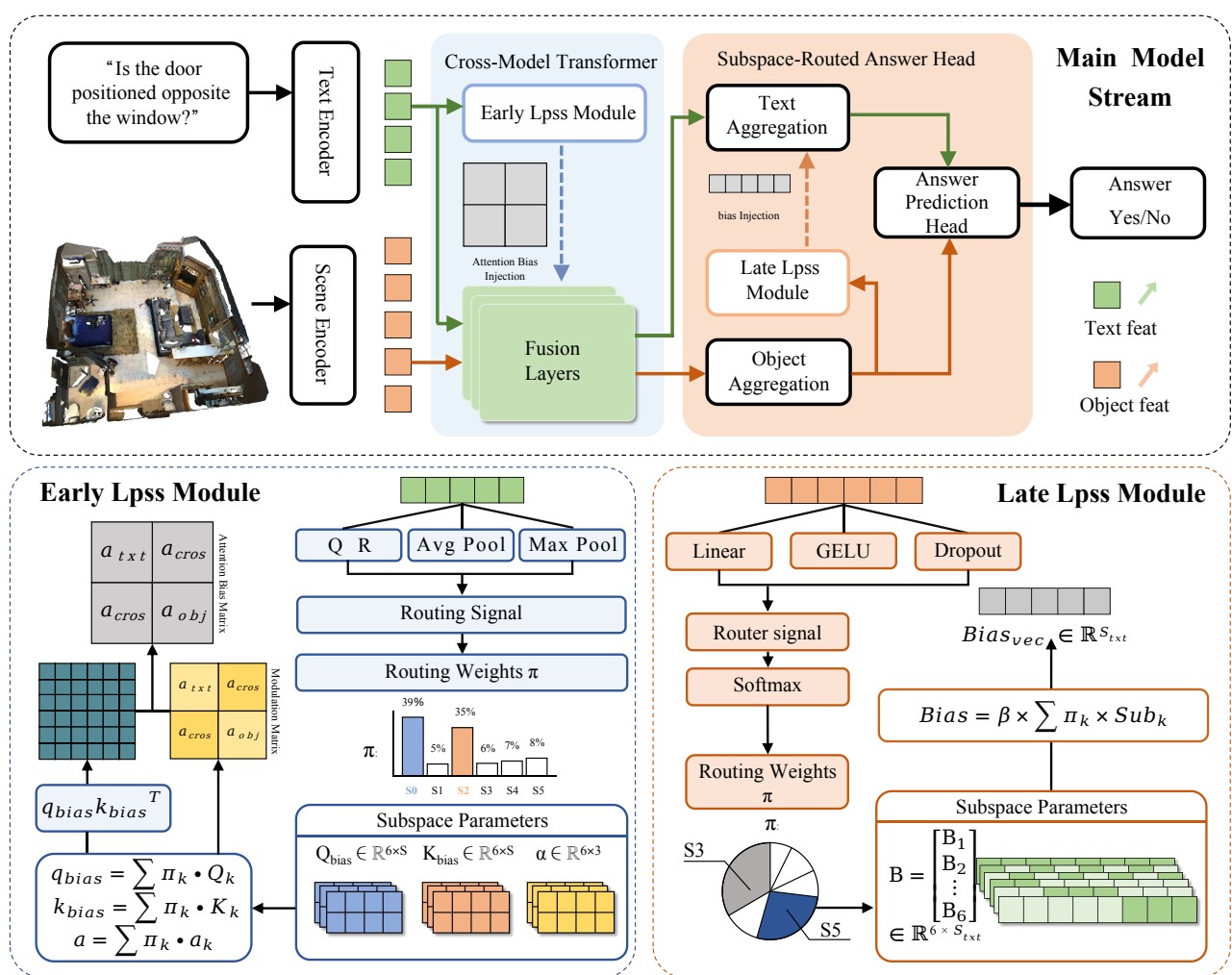

*Figure 2.* **Overview of the DualLPSS framework.** (a) **Early LPSS** utilizes the question representation as a routing signal to select from six learnable subspaces, generating a $T \times T$ attention bias matrix ($T = L + N$ is the total sequence length). This matrix is injected into the self-attention mechanism of the 0-th Transformer layer (pre-Softmax) to regulate cross-modal interaction directions. (b) **Late LPSS** employs the scene representation as a routing signal to select from six learnable bias vectors, producing a bias vector of length $L$. This vector is injected into the attention scores of the text aggregation module to guide focus toward critical question regions.

## 4. Method

### 4.1. Overview

Existing models falter in assertion verification, failing to accommodate requisite differentiated reasoning strategies and scene-dependent focus. In our comparative experiments, we observe that Negation and Spatial assertions yield consistently low accuracy across most baseline methods; the former requires parsing logical scope, correlating strongly with text understanding; the latter demands precise relationship reasoning, correlating strongly with scene comprehension. These findings motivate a dual-stage design targeting both modalities. To bridge this gap, we propose DualLPSS, which introduces type-aware routing at two pivotal stages. **Early LPSS** operates before cross-modal fusion, using question representations to inject attention biases and adapt *how* to fuse information based on question semantics.

**Late LPSS** operates after fusion, using scene representations to inject aggregation biases and adapt *what* to focus on based on scene content. The dual-stage injection positions and information flow are illustrated in Figure 2.

### 4.2. Encoders and Fusion Layer

The textual input is encoded by the first four layers of BERT (Devlin et al., 2019) (fine-tuned with learning rate multiplier 0.1) to obtain $\mathbf{T} \in \mathbb{R}^{B \times L \times D}$, where $B$ is the batch size, $L$ is the sequence length, and $D = 768$. Simultaneously, object point clouds are processed by Point-Net++ (Qi et al., 2017) to extract geometric features, followed by a spatial Transformer (4 layers, 12 heads) to model inter-object relationships, yielding $\mathbf{O} \in \mathbb{R}^{B \times N \times D}$, where $N$ denotes the number of objects. These features are concatenated along the sequence dimension to form a joint

sequence $\mathbf{J} = [\mathbf{T}; \mathbf{O}] \in \mathbb{R}^{B \times T \times D}$ ($T = L + N$ is the total sequence length). We employ a dedicated query to extract the question representation (QR). At each self-attention layer, the QR's query attends over the keys of all tokens to compute a sequence-level weighted sum, then updates through the residual and feed-forward blocks; stacking layers yields a QR that aggregates sentence-level context at every layer with dimension $D = 768$. This joint sequence is then fed into a 4-layer Transformer for cross-modal fusion, yielding fused representations $\mathbf{T}^{\text{out}} \in \mathbb{R}^{B \times L \times D}$ and $\mathbf{O}^{\text{out}} \in \mathbb{R}^{B \times N \times D}$. Specifically, the bias matrix $\mathbf{A}^{\text{early}} \in \mathbb{R}^{T \times T}$ generated by Early LPSS is injected into the self-attention calculation of the 0-th layer, while the bias vector $\mathbf{b}^{\text{late}} \in \mathbb{R}^{L}$ generated by Late LPSS is injected during the subsequent text aggregation phase.

### 4.3. Early LPSS

Early LPSS takes a routing signal $\mathbf{r}_q \in \mathbb{R}^{D}$ as input and outputs an attention bias matrix $\mathbf{A}^{\text{early}} \in \mathbb{R}^{T \times T}$, which is added to the $\mathbf{Q}\mathbf{K}^{\top}/\sqrt{d_k}$ term in the 0-th Transformer layer before Softmax, where $d_k = D/h = 64$ is the dimension per attention head with $h = 12$ heads. We inject at layer 0 to maximally influence early feature interaction. This module maintains $K = 6$ learnable subspaces: each subspace $k \in \{1, \ldots, K\}$ comprises a Query-side bias $\mathbf{p}_k^q \in \mathbb{R}^{T}$ (encoding positions that actively attend), a Key-side bias $\mathbf{p}_k^k \in \mathbb{R}^{T}$ (encoding positions to be attended), and three scalar structural strengths $\alpha_k^{\text{cross}}, \alpha_k^{\text{txt}}, \alpha_k^{\text{obj}}$ controlling attention gains for cross-modal, intra-text, and intra-object regions, respectively. Stacking across subspaces yields $\mathbf{P}^q \in \mathbb{R}^{K \times T}$, $\mathbf{P}^k \in \mathbb{R}^{K \times T}$, and $\mathbf{\Lambda} \in \mathbb{R}^{K \times 3}$ whose rows are $[\alpha_k^{\text{cross}}, \alpha_k^{\text{txt}}, \alpha_k^{\text{obj}}]$; the row-normalized versions $\tilde{\mathbf{P}}^q, \tilde{\mathbf{P}}^k, \tilde{\mathbf{\Lambda}}$ are used in the orthogonality term. Routing weights $\boldsymbol{\pi}^{\text{early}} \in \mathbb{R}^{K}$ are computed from $\mathbf{r}_q$ via a two-layer MLP with Gumbel-Softmax (Jang et al., 2017). We apply Soft Top-2 selection: the two largest weights are retained, while others are clamped to $\epsilon = 0.02$ rather than zero. This balances sparsity and gradient flow; hard selection would block gradients to inactive subspaces, while full Softmax dilutes routing signals. The aggregated subspace parameters are weighted sums:

$$\bar{\mathbf{p}}^q = \sum_k \pi_k^{\text{early}} \mathbf{p}_k^q, \tag{1}$$

$$\bar{\mathbf{p}}^k = \sum_k \pi_k^{\text{early}} \mathbf{p}_k^k, \tag{2}$$

$$\bar{\alpha}^* = \sum_k \pi_k^{\text{early}} \alpha_k^*. \tag{3}$$

The synthesized bias matrix is then defined as:

$$\begin{aligned} \mathbf{A}^{\text{early}}[i,j] = \beta^{\text{early}} \big( &\bar{\mathbf{p}}^q[i] + \bar{\mathbf{p}}^k[j] + \bar{\alpha}^{\text{cross}} \mathbf{C}[i,j] \\ &+ \bar{\alpha}^{\text{txt}} \mathbf{T}_m[i,j] + \bar{\alpha}^{\text{obj}} \mathbf{O}_m[i,j] \big) \end{aligned} \tag{4}$$

where $\beta^{\text{early}}$ is a global scaling factor, and $\mathbf{C}, \mathbf{T}_m, \mathbf{O}_m \in \{0,1\}^{T \times T}$ are binary masks defined as follows. We define: $\mathbf{T}_m[i,j] = 1$ iff $i, j \in [0, L)$ (intra-text); $\mathbf{O}_m[i,j] = 1$ iff

$i, j \in [L, T)$ (intra-object); $\mathbf{C}[i,j] = 1$ iff exactly one of $i, j$ is in $[0, L)$ (cross-modal).

### 4.4. Late LPSS

Late LPSS takes the scene representation $\mathbf{v} \in \mathbb{R}^{D'}$ ($D' = 512$) as input and outputs a bias vector $\mathbf{b}^{\text{late}} \in \mathbb{R}^{L}$ for injection into the text aggregation attention scores. Specifically, $\mathbf{v}$ is pooled by attention-based flattening (AttFlat) (Yu et al., 2019): a 2-layer MLP is applied to each fused object feature $\mathbf{o}_i^{\text{out}}$ to produce attention logits (glimpse = 1), followed by Softmax weighting and a linear projection to $D' = 512$. This module maintains $K = 6$ learnable bias vectors $\mathbf{B}_k \in \mathbb{R}^{L}$, each encoding an importance distribution over text positions (positive values enhance, negative values suppress). A router maps $\mathbf{v}$ to $K$-dimensional logits via a two-layer MLP, constrains them with $\tanh$ to $[-s, s]$ (we use $s = 5.0$), and computes routing weights $\boldsymbol{\pi}^{\text{late}}$ via Gumbel-Softmax during training. Initialization differs by stage: Early LPSS uses random orthogonal initialization to avoid preset fusion templates, whereas Late LPSS uses simple positional patterns only as a warm start to stabilize text aggregation. The final bias vector is a weighted sum:

$$\mathbf{b}^{\text{late}} = \beta^{\text{late}} \sum_{k=1}^{K} \pi_k^{\text{late}} \mathbf{B}_k \in \mathbb{R}^{L} \tag{5}$$

where $\beta^{\text{late}}$ is a global scaling factor. During aggregation, a learned query over $\mathbf{T}^{\text{out}}$ produces $\mathbf{z}^{\text{raw}} \in \mathbb{R}^{L}$; before Softmax, we add the bias as follows:

$$\mathbf{z}^{\text{txt}} = \mathbf{z}^{\text{raw}} + \mathbf{b}^{\text{late}}, \quad \mathbf{a}^{\text{txt}} = \text{Softmax}(\mathbf{z}^{\text{txt}}) \tag{6}$$

The final language representation is $\mathbf{t} = \sum_i a_i^{\text{txt}} \mathbf{t}_i^{\text{out}}$.

### 4.5. Dual-Stage Synergy

Neither stage alone suffices: Early LPSS cannot adapt to scene variations, while Late LPSS cannot influence how features are fused. Together, they enable end-to-end adaptive reasoning. The complete pipeline operates as follows: **(1)** The routing signal $\mathbf{r}_q$ is produced by an enhanced generator that combines the QR (first token of $\mathbf{T}$) with masked average and max pooling of $\mathbf{T}$, followed by a linear projection to $D$ dimensions, guiding Early LPSS to generate $\mathbf{A}^{\text{early}}$; **(2)** $\mathbf{J}$ is fused via a 4-layer Transformer with $\mathbf{A}^{\text{early}}$ injected at the 0-th layer, yielding $\mathbf{T}^{\text{out}}$ and $\mathbf{O}^{\text{out}}$; **(3)** Scene representation $\mathbf{v}$ is pooled from $\mathbf{O}^{\text{out}}$, guiding Late LPSS to produce $\mathbf{b}^{\text{late}}$; **(4)** $\mathbf{b}^{\text{late}}$ is injected during text aggregation to obtain $\mathbf{t}$. Finally, $\mathbf{t}$ is projected to $D'$ dimensions via a linear layer, then summed element-wise with $\mathbf{v}$ and passed through an MLP for binary prediction. Both stages use $K = 6$ subspaces (matching the number of semantic types; see Section 5.4 for sensitivity analysis) but maintain independent parameters.

*Table 2.* **Performance comparison with state-of-the-art methods on the 3DSAV test set.** The left block reports overall metrics: Accuracy (Acc.), Precision (Pre.), Recall (Rec.), and Macro-F1 (F1). The right block details accuracy across six semantic types. **Bold** denotes the best results, and underlined indicates second best.

| Type | Method | Venue | Overall (%) | | | | Per-Type Accuracy (%) | | | | | |
|---|---|---|---|---|---|---|---|---|---|---|---|---|
| | | | Acc. | Pre. | Rec. | F1 | Spa. | Neg. | Att. | Qua. | Exi. | Ver. |
| LLM[†] | 3D-LLM | NeurIPS'23 | 61.0 | 58.4 | 53.6 | 55.9 | 59.1 | 55.7 | 61.3 | 58.0 | 83.2 | 54.9 |
| | Chat-Scene | NeurIPS'24 | 80.5 | **79.7** | 77.5 | 78.6 | 73.9 | 69.8 | 79.8 | **95.3** | 93.7 | 79.8 |
| | Video3D-LLM | CVPR'25 | 70.3 | 70.4 | 69.6 | 69.6 | 62.7 | 73.6 | 68.4 | 76.4 | 80.1 | 64.4 |
| | LLaVA-3D | ICCV'25 | 72.1 | 74.2 | 60.6 | 66.7 | 69.8 | 65.2 | 68.2 | 74.6 | 91.0 | 72.9 |
| SM[‡] | ScanQA | CVPR'22 | 71.5 | 72.1 | 57.4 | 63.9 | 67.9 | 69.6 | 71.9 | 79.8 | 68.9 | 73.1 |
| | BUTD-DETR | ECCV'22 | 74.5 | 72.1 | 72.9 | 72.5 | 67.5 | 68.0 | 74.0 | 80.4 | 92.3 | 75.6 |
| | FE-3DGQA | NeurIPS'22 | 73.3 | 72.6 | 67.5 | 70.0 | 68.0 | 63.9 | 73.1 | 78.3 | 90.8 | 77.5 |
| | 3D-VisTA | ICCV'23 | 74.6 | 72.6 | 72.0 | 72.3 | 66.1 | 68.5 | 75.9 | 80.7 | 91.9 | 75.6 |
| | BridgeQA | AAAI'24 | 73.4 | 73.0 | 67.1 | 69.9 | 67.1 | 65.0 | 76.4 | 75.6 | 91.3 | 76.3 |
| | **Ours** | N/A | **82.5** | 79.2 | **84.3** | **81.7** | **76.9** | **77.2** | **81.3** | 90.2 | **94.4** | **82.6** |

[†] Large Language Models (LLMs) adapted via LoRA fine-tuning; [‡] Specialized Models (SMs) fully trained on 3DSAV.

Spa.=Spatial, Neg.=Negation, Att.=Attribute, Qua.=Quantity, Exi.=Existence, Ver.=Verification.

## 4.6. Training Objectives

The total loss comprises a primary classification loss and auxiliary losses for both stages:

$$\mathcal{L} = w_{\mathrm{ans}}\mathcal{L}_{\mathrm{ans}} + w_{\mathrm{aux}}(\mathcal{L}_{\mathrm{aux}}^{E} + \mathcal{L}_{\mathrm{aux}}^{L}) \tag{7}$$

For the first 25 epochs, Stage 1 uses $w_{\mathrm{ans}} = 1.0$ and $w_{\mathrm{aux}} = 1.0$; for the last 15 epochs, Stage 2 uses $w_{\mathrm{ans}} = 10.0$ and $w_{\mathrm{aux}} = 0.8$.

**Primary Loss.** $\mathcal{L}_{\mathrm{ans}}$ combines Focal Loss (Lin et al., 2017) with label smoothing ($\gamma = 2$, $\epsilon_s = 0.1$) to focus on hard examples while preventing overconfidence:

$$\mathcal{L}_{\mathrm{ans}} = -w_{\mathrm{FL}}(1 - p_y)^{\gamma} \sum_{c} \tilde{y}_c \log p_c \tag{8}$$

$$\tilde{y}_c = (1 - \epsilon_s)y_c + \frac{\epsilon_s}{C}$$

where $p_c$ is the predicted probability for class $c$, $y_c \in \{0, 1\}$ is the one-hot ground truth indicator, $w_{\mathrm{FL}}$ is the class balancing weight, $\gamma$ is the focusing parameter, $p_y$ is the predicted probability of the ground-truth class, $\epsilon_s$ is the label smoothing factor, and $C = 2$ is the number of classes.

**Early LPSS Auxiliary Loss.** $\mathcal{L}_{\mathrm{aux}}^{E}$ includes orthogonality, entropy regularization, and diversity terms:

$$\mathcal{L}_{\mathrm{aux}}^{E} = \lambda_{\mathrm{ortho}}^{E}\mathcal{L}_{\mathrm{ortho}}^{E} + \lambda_{\mathrm{ent}}^{E}\mathcal{L}_{\mathrm{ent}}^{E} + \lambda_{\mathrm{div}}^{E}\mathcal{L}_{\mathrm{div}}^{E} \tag{9}$$

Orthogonality enforces perpendicularity among Query/Key biases and structural parameters ($\tilde{\mathbf{P}}^q, \tilde{\mathbf{P}}^k, \tilde{\mathbf{\Lambda}}$ are row-normalized matrices); entropy regularization minimizes the gap to maximum entropy ($\log K$), thereby promoting routing diversity; diversity loss encourages distinct cross-modal configurations $\boldsymbol{\alpha}^{\mathrm{cross}}$. Here $\tilde{\mathbf{\Lambda}}$ is the row-normalized

structural-strength matrix defined in Early LPSS:

$$\mathcal{L}_{\mathrm{ortho}}^{E} = \frac{1}{3} \sum_{\mathbf{X} \in \{\tilde{\mathbf{P}}^q, \tilde{\mathbf{P}}^k, \tilde{\mathbf{\Lambda}}\}} \|\mathbf{X}\mathbf{X}^{\top} - \mathbf{I}\|_F \tag{10}$$

$$\mathcal{L}_{\mathrm{ent}}^{E} = \log K + \frac{1}{B} \sum_{i=1}^{B} \sum_{k} \pi_{ik}^{E} \log \pi_{ik}^{E} \tag{11}$$

$$\mathcal{L}_{\mathrm{div}}^{E} = -\mathrm{Var}(\boldsymbol{\alpha}^{\mathrm{cross}}) \tag{12}$$

**Late LPSS Auxiliary Loss.** $\mathcal{L}_{\mathrm{aux}}^{L}$ includes load balancing, orthogonality, and entropy terms:

$$\mathcal{L}_{\mathrm{aux}}^{L} = \lambda_{\mathrm{lb}}^{L}\mathcal{L}_{\mathrm{lb}}^{L} + \lambda_{\mathrm{ortho}}^{L}\mathcal{L}_{\mathrm{ortho}}^{L} + \lambda_{\mathrm{ent}}^{L}\mathcal{L}_{\mathrm{ent}}^{L} \tag{13}$$

Load balancing penalizes deviations from a uniform subspace usage; orthogonality ensures bias vectors $\tilde{\mathbf{B}}$ are distinct; entropy regularization encourages diversity:

$$\mathcal{L}_{\mathrm{lb}}^{L} = \sum_{k=1}^{K} \left( \bar{\pi}_k^{L} - \frac{1}{K} \right)^2 \tag{14}$$

$$\mathcal{L}_{\mathrm{ortho}}^{L} = \|\tilde{\mathbf{B}}\tilde{\mathbf{B}}^{\top} - \mathbf{I}\|_F \tag{15}$$

$$\mathcal{L}_{\mathrm{ent}}^{L} = \begin{cases} m(H_{\min} - H^L) & H^L < H_{\min} \\ (H_{\mathrm{tgt}} - H^L) & H_{\min} \leq H^L < H_{\mathrm{tgt}} \\ 0 & H^L \geq H_{\mathrm{tgt}} \end{cases} \tag{16}$$

where $\bar{\pi}_k^{L} = \frac{1}{B} \sum_{i=1}^{B} \pi_{ik}^{L}$ is the batch-averaged routing weight for subspace $k$, and $H^L = -\frac{1}{B} \sum_{i=1}^{B} \sum_{k} \pi_{ik}^{L} \log \pi_{ik}^{L}$ is the batch-averaged routing entropy (maximum $\log K \approx 1.79$ for $K = 6$), with $H_{\min} = 0.5$, $H_{\mathrm{tgt}} = 0.8$, and $m = 5.0$.

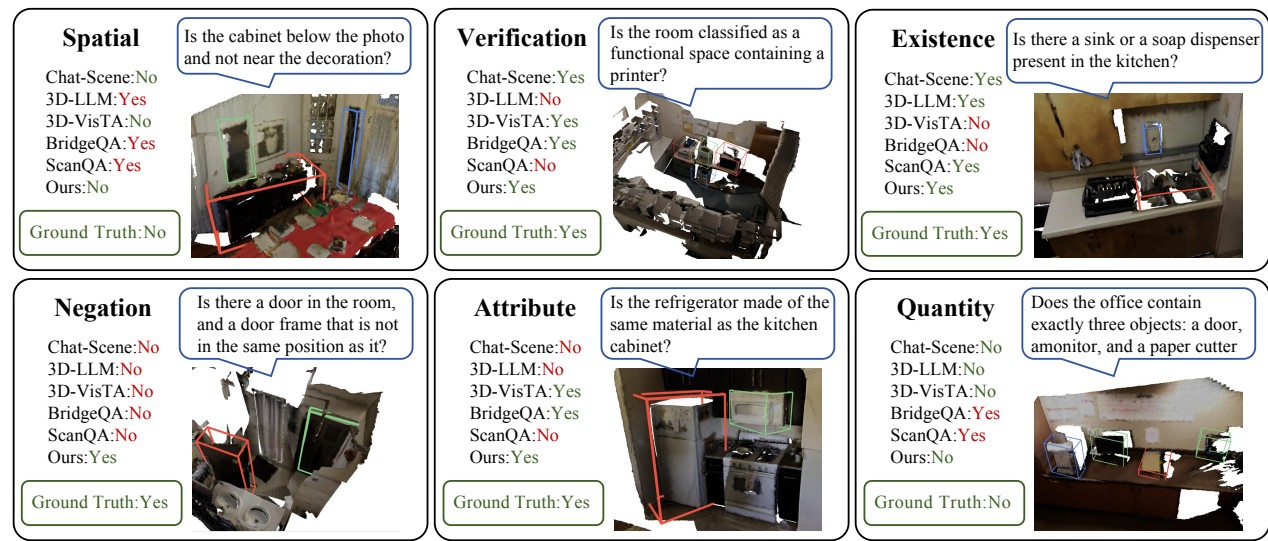

*Figure 3.* **Qualitative comparison across six semantic types.** DualLPSS correctly predicts all cases, while baseline methods make errors on different assertion types, particularly on complex negation reasoning.

## 5. Experiments

### 5.1. Implementation Details

**DualLPSS Training.** Both LPSS stages use $K = 6$ subspaces with Soft Top-2 selection to ensure gradient flow. Training uses Focal Loss ($\gamma = 2.0$, label smoothing 0.1), batch size 128, learning rate $1 \times 10^{-4}$, AdamW optimizer, and 40 epochs. A staged training strategy is applied with adjusted loss weights. The full training configuration is listed in Appendix B.

**Baseline Training.** LLM baselines (e.g., Chat-Scene) are fine-tuned via LoRA for 3 epochs. Specialized models (e.g., 3D-VisTA, BridgeQA) are adapted from their original implementations with task-specific modifications and fully trained on 3DSAV. See Appendix E for details.

### 5.2. Quantitative Analysis

Table 2 presents the comparison between DualLPSS and nine baselines. DualLPSS achieves 82.5% accuracy and 81.7% F1, outperforming all methods. DualLPSS ranks first in five out of six semantic types. The largest gains are on Negation (+3.6 pp) and Spatial (+3.0 pp).

We analyze baselines in two groups: Large Language Models (LLMs) and Specialized Models (SMs). LLMs like Chat-Scene benefit from large-scale pretraining but have high inference latency due to their size. SMs like 3D-VisTA are faster but use uniform fusion strategies without adapting to question types, leading to poor performance on certain types (e.g., ScanQA achieves only 68.9% on Existence). As shown in Table 3, DualLPSS balances efficiency and accuracy: training takes 6.3 min / 5.6GB per epoch, evaluation takes 22s / 3.8GB, comparable to SMs. Combined with

*Table 3.* **Efficiency and cross-dataset generalization.** LLM training cost is significantly higher, so the efficiency comparison focuses on specialized models. Train time (min) and peak GPU memory (GB) are per epoch; Eval time (s) and memory (GB) are on the test set. **Bold**/underline: best/second-best.

| Model | Train (min / GB) | Eval (s / GB) | ScanQA (%) | SQA3D (%) |
|---|---|---|---|---|
| 3D-LLM | N/A | N/A | 50.7 | 52.3 |
| Chat-Scene | N/A | N/A | 70.9 | 70.7 |
| Video3D-LLM | N/A | N/A | 63.4 | 61.6 |
| LLaVA-3D | N/A | N/A | 55.1 | 59.1 |
| ScanQA | 8.5 / 9.2 | 42 / 1.6 | 47.6 | 50.2 |
| BUTD-DETR | 12.4 / 11.4 | 8.1 / 2.5 | 58.5 | 64.0 |
| FE-3DGQA | 4.2 / 5.4 | 113 / 2.9 | 55.4 | 61.7 |
| 3D-VisTA | 9.0 / 6.7 | 1.1 / 0.2 | 60.2 | 65.5 |
| BridgeQA | 10.5 / 8.0 | 58 / 4.8 | 68.1 | 69.2 |
| **Ours** | 6.3 / 5.6 | 22 / 3.8 | **82.1** | **81.9** |

the 82.5% accuracy reported in Table 2, this demonstrates state-of-the-art performance with practical efficiency. This advantage comes from the dual-stage routing design: Early LPSS uses a 1D decomposition plus a lightweight router and routing-signal MLP, while Late LPSS uses a 512→256→6 router plus $6 \times 50$ bias parameters (about 133K parameters), enabling type-aware reasoning without the latency of LLMs.

**Cross-Dataset Generalization.** To test generalization, we construct evaluation subsets from ScanQA and SQA3D. Since these datasets use open-ended QA format, we select color queries ("What color is X?") and counting queries ("How many X?"), then convert them to verification format ("Is X [color]?" and "Are there N X?"). Ground-truth labels are determined by answer matching (see Appendix F for conversion rules). On these color/quantity subsets, DualLPSS reaches 82.1% and 81.9%, outperforming the next-

best Chat-Scene (70.9%, 70.7%) by double-digit margins. The gains remain after format conversion.

**Robustness and Transfer Checks.** Appendix C.2 reports checks under predicted object proposals, 2D keyframe baselines, harder spatial/negation conversions, outdoor scenes, and shortcut controls. Together, these settings check whether the gains rely on clean 3D inputs, 2D views, in-domain scenes, or simple lexical cues.

### 5.3. Qualitative Analysis

Figure 3 compares predictions of DualLPSS and baselines on six semantic types. DualLPSS correctly predicts all six cases, while all baselines make at least one incorrect prediction. The most notable finding is on the Negation type: for a complex assertion with nested logic, DualLPSS is the only method that predicts correctly; all compared baselines shown in Figure 3 predict incorrectly. This confirms the improvement on Negation shown in Table 2, where DualLPSS surpasses the strongest baseline Video3D-LLM by 3.6 percentage points. Negation assertions require parsing logical scope within text before matching with the scene; Early LPSS routes to enhance intra-text attention. In contrast, generative LLMs are trained for fluent outputs and may answer with descriptions rather than binary logical judgments.

### 5.4. Ablation Study

*Table 4.* **Ablation study on DualLPSS components.** Stage contributions, structural strengths, and routing mechanisms.

| Configuration | Acc. | Pre. | Rec. | F1 |
|---|---|---|---|---|
| *Stage Contributions* | | | | |
| w/o DualLPSS | 77.3 | 74.3 | 77.5 | 75.9 |
| + Early only | 81.9 | 79.6 | 81.8 | 80.7 |
| + Late only | 82.0 | 79.3 | 82.6 | 80.9 |
| *Structural Strengths* | | | | |
| w/o Text | 81.7 | 79.1 | 82.1 | 80.6 |
| w/o Cross | 82.0 | 79.2 | 82.8 | 81.0 |
| w/o Object | 82.2 | 79.7 | 82.5 | 81.1 |
| *Routing Mechanisms* | | | | |
| Hard Top-K | 81.2 | 78.5 | 81.7 | 80.1 |
| w/o Gumbel | 81.7 | 79.1 | 82.1 | 80.6 |
| **Full DualLPSS** | **82.5** | **79.2** | **84.3** | **81.7** |

**Component Analysis of DualLPSS.** Table 4 shows the contributions of Early and Late LPSS. The backbone without DualLPSS yields 77.3% accuracy. Early LPSS alone improves accuracy by 4.6%, and Late LPSS alone by 4.7%. Full DualLPSS reaches 82.5% (+5.2%) and adds 0.5 points over the stronger single-stage variant. This supports using the full two-stage design.

**Structural Strengths.** Removing intra-text strength $\alpha^{\text{txt}}$ causes the largest drop (-0.8%), as assertions require understanding internal logical structures before scene matching.

Cross-modal strength $\alpha^{\text{cross}}$ follows (-0.5%), while intra-object $\alpha^{\text{obj}}$ has minimal impact (-0.3%) since spatial relationships are already captured by the point cloud encoder.

**Routing Mechanism.** Replacing Gumbel-Softmax with standard Softmax drops accuracy by 0.8%, as Gumbel noise enhances gradient signals for routing. Hard Top-K causes a 1.3% drop by blocking gradients to inactive subspaces; with a 0.02 floor, Soft Top-K keeps inactive subspaces updated.

**Sensitivity to Subspace Count $K$.** We find that $K = 6$ achieves the optimal balance, outperforming -DualLPSS by 5.2%. Analysis shows Early LPSS is more sensitive to $K$ (dropping 1.3% at $K = 8$) than Late LPSS (dropping 1.0%), as it relies on semantic routing regularities. Detailed ablation results on $K$ are provided in Appendix C.

**Visualization of Routing Behavior** Figure 4 shows the activation distribution across subspaces. Early LPSS exhibits clear type-to-subspace correspondence: *Spatial* activates Subspace 0; *Existence* activates Subspace 2; *Negation* activates Subspace 1; *Quantity* activates Subspace 3. Late LPSS shows a different pattern: the peak subspaces are *not aligned* with question types. This misalignment confirms that Late LPSS routing is driven by scene content, achieving complementary specialization with Early LPSS. Semantic type labels are used only for dataset analysis and per-type evaluation, not as model inputs during training or inference. The correspondence in Figure 4 is therefore learned through the routing objective rather than imposed by type supervision.

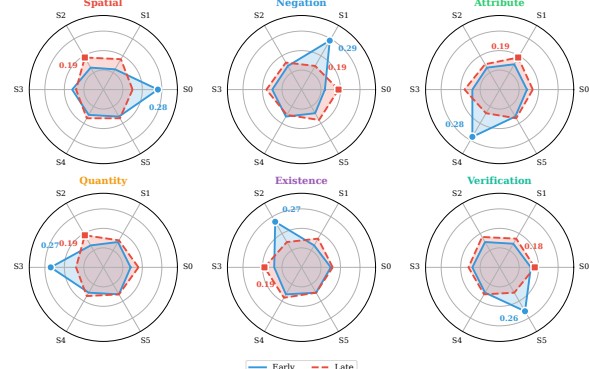

*Figure 4.* **Routing behavior.** Early and Late LPSS show type-specific and scene-dependent activation patterns, respectively.

## 6. Conclusion

This paper introduces 3D Scene Assertion Verification as a deterministic alternative to open-ended 3D-VQA for truth-value judgment. We construct **3DSAV**, a 22,520-sample benchmark over 682 ScanNet scenes, and evaluate existing 3D-VQA and 2D-VLM baselines under the same binary-verification protocol. We also propose **DualLPSS**, which combines early type-sensitive routing with late scene-guided text focus and outperforms the strongest baseline on the main benchmark and robustness checks. The remaining

errors concentrate in spatial and negated assertions, pointing to geometric grounding and ambiguity reduction as the main directions for future work.

**Limitation & Future Work.** Our formulation establishes a clean starting point; the remaining failure modes surfaced in the error analysis (Section D), notably spatial negation and fine-grained attributes, highlight the need to integrate interactive view acquisition to reduce ambiguity before verification. We plan to release standardized protocols for these extensions so that community baselines remain comparable as the task scope grows.

## Impact Statement

This paper presents work whose goal is to advance the field of Machine Learning by introducing a novel task and benchmark for 3D scene assertion verification. The primary applications are in robotics and embodied AI, where deterministic judgments are essential for safe and reliable decision-making. The dataset is constructed from publicly available ScanNet scenes with human verification to ensure quality. We do not foresee specific negative societal consequences that must be highlighted here.

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

# A. Dataset Construction Details

## A.1. LLM Generation Configuration

We use Qwen3-VL-Plus for automatic assertion generation. The generation process employs a sampling strategy with temperature 0.7 and Top-p 0.9, with a maximum generation length of 4096 tokens and output format strictly constrained to JSON arrays. Upon generation failure, the system applies a progressive retry strategy with up to 4 attempts, gradually strengthening prompt constraints to improve success rate.

The generation process uses 42 quality constraint rules to ensure that generated assertions satisfy requirements for semantic clarity, logical consistency, and format compliance. These constraints are divided into four categories: question constraints (1 to 15), answer constraints (16 to 25), logic constraints (26 to 35), and format constraints (36 to 42). The complete 42 constraint rules are detailed in Section G.

## A.2. Semantic Type Definitions and Classification Rules

### A.2.1. CLASSIFICATION DESIGN RATIONALE

Different assertion types require different core capabilities (e.g., spatial reasoning, negation understanding, quantity counting). A unified overall accuracy cannot reveal model strengths and weaknesses on specific reasoning dimensions. Therefore, we categorize assertions into six semantic types to support fine-grained diagnostic evaluation, helping researchers precisely identify model capability gaps.

### A.2.2. TYPE DEFINITIONS AND CLASSIFICATION RULES

Each assertion is assigned to a unique type through a priority-based rule matching system. **Priority** means that when an assertion matches multiple type patterns simultaneously (e.g., containing both the negation word "not" and the spatial word "near"), the system checks types from highest to lowest priority and stops upon the first match, ensuring each sample is assigned to exactly one type. The priority design follows the principle of "specific before general": types requiring special logical processing such as negation and quantification have the highest priority, while verification serves as a fallback category with the lowest priority. Rules are applied in the following order:

**(1) Negation** (Priority 1): Assertions containing explicit negation operators, requiring understanding of negation scope before verification. Keywords include "not", "n't", "no", "without", covering 99.2% of all Negation assertions. Example: "Is the bookshelf *not* next to any chair?"

**(2) Quantity** (Priority 2): Assertions involving numerical verification with explicit quantifiers. Keywords include "at least" (60.8%), "more than" (17.2%), "exactly" (10.4%), "fewer than", "at most". Example: "Are there *at least* 2 chairs *or at least* 3 tables?"

**(3) Existence** (Priority 3): Assertions querying object presence or absence. Keywords include "is there"/"are there" (64.5%), "contain" (24.0%), "have"/"has" (7.6%). Example: "*Is there* a picture hanging on the wall?"

**(4) Attribute** (Priority 4): Assertions about observable visual properties of objects. Color keywords (40.9%) include "color", "colored", "white", "black", "brown", "red", "blue", "green", "light", "dark"; material keywords (39.7%) include "wooden", "metal", "plastic", "glass", "made of"; state and visibility keywords (19.4%) include "open", "closed", "on", "off", "occluded", "visible". Example: "Is the armchair *white*?"

**(5) Spatial** (Priority 5): Assertions involving spatial relationships between objects. Keywords (87.1% coverage) include "near" (25.8%), "next to" (18.2%), "above" (12.6%), "under" (6.7%), "opposite" (5.4%), "left"/"right" (5.0%), "in front of", "behind", "between", "inside". Example: "Is the lamp located *above* the table?"

**(6) Verification** (Priority 6): Comprehensive assertions that do not match any of the above patterns, requiring holistic scene understanding. These samples have no dominant keywords and require integration of multiple scene aspects. Example: "Is the doorframe the only non-door object?"

## A.3. Human Verification Process

Human verification was completed independently by 8 annotators with 3D computer vision backgrounds, from diverse genders, ages, and educational backgrounds to ensure diverse evaluation perspectives. This diversity helps reduce individual bias and improve annotation reliability. At least 2 annotators reviewed each sample independently, and samples with disagreement were discarded.

To ensure annotators could objectively evaluate assertion validity, we provided comprehensive training materials covering the following evaluation dimensions: (1) Assertion-scene correspondence: whether objects mentioned in the assertion are visible in the scene; (2) Spatial relationship accuracy: whether spatial positions described match the scene; (3) Object attribute correctness: whether color, material, and size descriptions are accurate; (4) Logical consistency: whether internal logic is coherent and negation scope is clear. Training materials included detailed explanations with positive and negative examples.

To ensure consistency and objectivity, we employed cross-validation across annotators. Inter-annotator agreement was continuously monitored during annotation using quadratic weighted agreement, with an average score of 0.847 that exceeds the commonly accepted "substantial" threshold of 0.8; raw agreement reached approximately 92.3%.

Annotation was completed through a custom 3D visualization web interface supporting the following tasks: ground truth correction (verifying and correcting Yes/No labels),

sample cleaning (filtering invalid samples), and grounding annotation (selecting relevant object identifiers supporting the judgment), as shown in Figure A1.

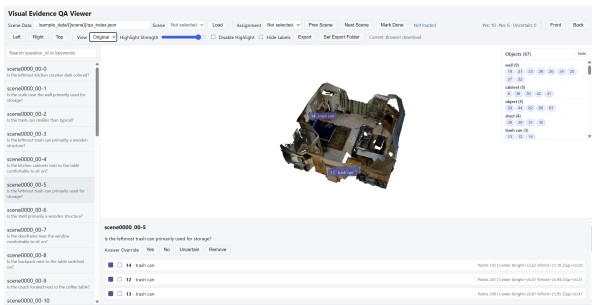

*Figure A1.* **3D visualization verification interface.**

## A.4. Annotation Guidelines

Table A1 lists the verification guidelines.

*Table A1.* **Detailed Annotation Guidelines.**

| Task | Guidelines |
|------|-----------|
| Ground Truth Correction | 1. Read assertion and understand semantics; 2. Locate objects in 3D scene; 3. Judge truth value; 4. Correct label if needed |
| Sample Cleaning | Filter invalid samples: 1. Visual hallucination (non-existent objects); 2. External knowledge dependency; 3. Semantic ambiguity |
| Grounding Annotation | 1. Identify objects in assertion; 2. Select object IDs; 3. Ensure objects support the judgment conclusion |

## A.5. Data Statistics

This section provides detailed statistics of the 3DSAV benchmark, including data splits, semantic type distribution, label balance analysis, and logical structure characteristics.

### A.5.1. DATA SPLITS AND LABEL DISTRIBUTION

*Table A2.* **3DSAV Dataset Split Statistics.**

| Split | Samples | Scenes | Yes (%) | No (%) |
|-------|---------|--------|---------|--------|
| Train | 18,620 | 547 | 48.4 | 51.6 |
| Val | 2,001 | 68 | 46.2 | 53.8 |
| Test | 1,899 | 67 | 46.1 | 53.9 |
| **Total** | **22,520** | **682** | **48.0** | **52.0** |

The dataset maintains approximately balanced label distribution across all splits, with roughly 48% positive samples (Yes) and 52% negative samples (No). This balance mitigates potential bias from class imbalance and ensures fairness in training and evaluation.

### A.5.2. SEMANTIC TYPE DISTRIBUTION ANALYSIS

*Table A3.* **Semantic Type Distribution and Yes/No Balance.**

| Type | Train | Val | Test | Yes | No |
|------|-------|-----|------|-----|-----|
| Spatial | 24.5 | 26.6 | 24.7 | 44.7 | 55.3 |
| Negation | 19.8 | 16.6 | 19.6 | 46.5 | 53.5 |
| Attribute | 18.4 | 18.3 | 18.5 | 44.4 | 55.6 |
| Quantity | 16.5 | 17.5 | 16.6 | 38.4 | 61.6 |
| Existence | 11.5 | 11.6 | 11.4 | 84.7 | 15.3 |
| Verification | 9.3 | 9.4 | 9.2 | 38.7 | 61.3 |

Notably, the *Existence* type exhibits a significant positive sample bias (84.7% Yes), reflecting that most existence assertions query objects that are indeed present in the scene. In contrast, *Quantity* and *Verification* show more negative samples because exact counts and comprehensive criteria are harder to satisfy.

### A.5.3. LOGICAL STRUCTURE STATISTICS

*Table A4.* **Logical Structure (Overall Distribution).**

| Structure | Count | Proportion |
|-----------|-------|-----------|
| Negation Operator | 4,385 | 19.5% |
| Quantifier | 3,613 | 16.0% |
| Disjunction (OR) | 2,256 | 10.0% |
| Conjunction (AND) | 732 | 3.3% |

*Note on taxonomies:* The "Quantifier" proportion (16.0%) reported here follows the logical-structure view, while the main paper's "Quantitative" proportion (16.6%) follows the semantic-type taxonomy. The two percentages differ because the grouping rules are different.

*Table A5.* **Logical Structure (Proportion by Semantic Type, %).**

| Type | OR | AND | Neg. | Quant. |
|------|-----|-----|------|--------|
| Quantity | 52.1 | 0.9 | 0.0 | 93.6 |
| Negation | 1.1 | 2.3 | 99.2 | 0.2 |
| Existence | 4.7 | 11.8 | 0.0 | 0.5 |
| Verification | 1.5 | 2.1 | 0.0 | 2.9 |
| Attribute | 1.3 | 1.6 | 0.0 | 0.4 |
| Spatial | 0.9 | 3.3 | 0.0 | 0.1 |

Logical structure analysis reveals clear type specialization: *Quantity* assertions primarily contain quantifiers (93.6%) and disjunctive structures (52.1%), while *Negation* assertions almost exclusively contain negation operators (99.2%). These patterns support the taxonomy.

## A.5.4. ASSERTION COMPLEXITY AND OBJECT STATISTICS

*Table A6.* **Assertion Length Statistics.**

| Overall Length | | | By Type (Words) | | |
|---|---|---|---|---|---|
| Stat. | Chars | Words | Type | Mean | Std |
| Mean | 43.2 | 8.5 | Quantity | 11.8 | 3.2 |
| Std Dev | 12.4 | 2.5 | Spatial | 8.4 | 1.6 |
| Min | 20 | 5 | Negation | 8.3 | 1.6 |
| Max | 105 | 19 | Existence | 7.8 | 1.5 |
| Median | 41.0 | 8.0 | Verification | 7.6 | 1.3 |
| 95th %ile | 70.0 | 14.0 | Attribute | 6.6 | 1.4 |

*Table A7.* **Object Reference Statistics.**

| Object Reference | | Distribution | |
|---|---|---|---|
| Statistic | Value | Type | Share |
| Avg. objects/assertion | 3.20 | Single-obj | 31.8% |
| Category vocabulary | 489 types | Two-obj | 31.4% |
| Max objects | 95 | Multi-obj | 36.7% |

The 95th percentile (95th %ile) indicates that 95% of assertions contain at most 70 characters (or 14 words). *Quantity* assertions are notably longer (average 11.8 words) due to compound conditions such as "at least X or at least Y". *Attribute* assertions are shortest (6.6 words), typically following a simple pattern like "Is the [object] [color]?" Over two-thirds of assertions (68.1%) involve multiple objects, requiring models to perform complex inter-object relationship reasoning. The most frequent categories are *chair* (19,169 occurrences), *table* (4,933), and *window* (3,669).

## A.6. Evaluation Metrics

Primary evaluation metrics include Accuracy, Precision, Recall, and Macro-F1. Additionally, type-specific accuracy $\text{Acc}_t = C_t/N_t$ is used for diagnostic analysis, where $C_t$ is the number of correct predictions for type $t$ and $N_t$ is the total number of samples of type $t$. The Delta metric $\Delta_t = \text{Acc}_{\text{overall}} - \text{Acc}_t$ measures deviation from overall performance for each type, where positive values indicate greater difficulty and negative values indicate relative ease.

## A.7. Prompt Design

Assertion generation employs a two-stage structure combining a system prompt and a scene prompt. The system prompt defines the generation task and quality constraints (Figure A2), while the scene prompt provides specific object catalogs and generation plans for each scene (Figure A3).

**Task:** Generate high-quality Yes/No assertion QA pairs for 3D indoor scenes.
**Input:** Scene metadata (scene_id, category), object catalog (id → label mapping), allowed label vocabulary.
**Output:** JSON array, each item containing: `scene_id`, `question`, `answer` ∈ {Yes, No}, `type`, `object_labels`, `object_ids`.
**Instructions:**

1. Question length must be between 10 to 200 characters.
2. Must begin with an interrogative word or end with a question mark.
3. Vague pronouns (this/that/it) are prohibited.
4. Must reference specific object names from the object catalog.
5. Spatial relations must use standard directional terms.
   ⋮
42. Field naming must follow consistent conventions.

**Notice:** All labels in `object_labels` must appear verbatim in the question text.
**Note:** Questions requiring external knowledge (e.g., "Is this a famous painting?") are invalid.

*Figure A2.* **System prompt for assertion generation.**

**Task:** Generate assertion QA pairs of specified quantity and distribution based on scene information.
**Input:**

- Scene ID: scene0481_00, Predicted category: bedroom
- Target generation count: 35 QA pairs
- Object catalog: {0: wall, 1: floor, 5: bed, 8: nightstand, ...}

**Output:** JSON array generated strictly following slot order.
**Instructions:**

1. Select object labels to use from the allowed label vocabulary.
2. Write question text referencing only selected labels.
3. Self-check: every label in the question must appear in `object_labels`.
4. Generate each semantic type according to target distribution.
5. Answer distribution follows Yes:No ≈ 48:52.

**Notice:** Generation count must exactly equal the target count.

*Figure A3.* **Scene prompt example.**

Few-shot examples are provided in the format shown in Table A8 to guide output structure.

*Table A8.* **Few-shot Examples.**

| Question | Ans. | Type | Objects |
|---|---|---|---|
| Is the wooden chair next to the table? | Yes | Spatial | chair, table |
| Does the room contain a refrigerator or microwave? | No | Exist. | refrig., micro. |

# B. DualLPSS Model Implementation Details

## B.1. Overall Architecture

The DualLPSS (Dual-stage Layered Progressive Subspace Specialization) framework introduces a dual-stage subspace routing mechanism on top of the 3D-VisTA base architecture. The text encoder uses the first 4 layers of BERT-base with hidden dimension 768 and input sequence length of 50 tokens. The point cloud encoder uses PointNet++ as the backbone with hidden dimension 768 and ground-truth point clouds. The unified encoder consists of 4 Transformer layers, each with 12 attention heads and hidden dimension 768. The QA head uses AttFlat with a 256-dimensional MLP to produce 512-dimensional features, followed by a $512 \rightarrow 768 \rightarrow 2$ classifier with dropout rate 0.3.

## B.2. Early LPSS Module

Early LPSS injects attention masks before layer 0 of the unified encoder to guide cross-modal fusion direction. The module contains $K = 6$ subspaces, each maintaining a Query-side bias $\mathbf{p}_k^q \in \mathbb{R}^T$, a Key-side bias $\mathbf{p}_k^k \in \mathbb{R}^T$ (where $T = L + N$ is the total sequence length), and three structural strengths $\alpha_k^{\text{cross}}, \alpha_k^{\text{txt}}, \alpha_k^{\text{obj}}$ for cross-modal, intra-text, and intra-object regions, respectively. The router is a two-layer MLP mapping the QR representation (EnhancedRoutingSignal over QR token + masked avg/max pooling of $\mathbf{T}$) to $K$ routing logits. Routing temperature $\tau$ is set to 0.5, using Gumbel-Softmax with Gumbel temperature 1.0. Top-K is set to 2 with soft selection retaining minimum weight 0.02 to ensure gradient flow through all subspaces. Bias scaling starts at 12.0, and gradient scaling is set to 10.0.

Query/key biases are randomly initialized and orthogonalized via QR; structural strengths are initialized with uniform random values (not orthogonalized), without fixed semantic templates or baseline subspaces; the routing objective learns this correspondence, and no type-label supervision is used.

## B.3. Late LPSS Module

Late LPSS injects attention bias in the AttFlat layer of the QA classification head for refined answer prediction. The module also contains $K = 6$ subspaces, each maintaining an attention bias pattern of length 50. The router maps 512-dimensional object_feat to $K$-dimensional logits via a two-layer MLP ($512 \rightarrow 256 \rightarrow K$), constrains them to $[-5, 5]$ via $\texttt{tanh}$, and uses Gumbel-Softmax during training (Softmax at test time). Routing temperature is set to 1.5, with Top-K=2 and bias scaling initialized to 5.0.

Subspace bias pattern initialization uses semantic design: Subspace 0 attends to the first 5 tokens at sequence beginning (question start), Subspace 1 attends to the last 10 tokens at sequence end (question end), Subspace 2 attends to middle tokens 20-30 (supplementary information), Subspace 3 uses uniform distribution as a general fallback, and Sub-

spaces 4 and 5 attend to even and odd positions respectively for structured attention. All patterns are L2-normalized so each subspace has comparable influence.

## B.4. Auxiliary Loss Functions

Multiple auxiliary loss functions are used during training to promote subspace differentiation. Let $K$ be the total number of subspaces and $B$ be the batch size.

**Orthogonality Loss** enforces mutual orthogonality among subspace patterns. For Early LPSS, we apply orthogonality across normalized query biases, key biases, and structural strengths; for Late LPSS, we apply orthogonality to the normalized subspace bias matrix $\mathbf{B} \in \mathbb{R}^{K \times L}$ via $\mathcal{L}_{\text{ortho}} = \|\mathbf{B}\mathbf{B}^\top - \mathbf{I}\|_F$.

**Load Balancing Loss** penalizes imbalanced subspace usage within a batch. Let $u_k$ be the average usage rate of subspace $k$ in the current batch (i.e., batch average of routing weights). The loss is computed as $\mathcal{L}_{\text{lb}} = \sum_{k=1}^{K}(u_k - 1/K)^2$. This loss ensures all subspaces are adequately utilized, preventing routing collapse to a few subspaces.

**Entropy Regularization Loss** controls the concentration of routing distribution using per-sample entropy $H = -\frac{1}{B}\sum_{i=1}^{B}\sum_k \pi_{ik} \log \pi_{ik}$. For Late LPSS, we apply a target-based penalty with $H_{\min} = 0.5$, $H_{\text{tgt}} = 0.8$, and multiplier $m = 5.0$.

Auxiliary loss weights for Early LPSS are: orthogonality $\lambda_{\text{ortho}}^E = 0.05$, entropy regularization $\lambda_{\text{ent}}^E = 0.1$, diversity $\lambda_{\text{div}}^E = 0.02$. Auxiliary loss weights for Late LPSS are: orthogonality $\lambda_{\text{ortho}}^L = 0.5$, load balancing $\lambda_{\text{lb}}^L = 0.3$, entropy regularization $\lambda_{\text{ent}}^L = 0.1$.

## B.5. Training Configuration

Training uses batch size 128 with AdamW optimizer ($\beta_1 = 0.9$, $\beta_2 = 0.98$). Base learning rate is $1 \times 10^{-4}$, with BERT learning rate multiplier 0.1, Late LPSS learning rate multiplier 10.0, and Early LPSS learning rate multiplier 5.0. Gradient clipping threshold is 5.0, total training epochs is 40, warmup steps is 500, and random seed is 42.

The main loss function uses Focal Loss with label smoothing, with Focal $\gamma$ set to 2.0, class balancing weight $w_{\text{FL}}$ set to 1.0, and smoothing coefficient 0.1. Training follows a staged strategy: Stage 1 (first 25 epochs) uses answer loss weight 1.0 and auxiliary loss weight 1.0 to establish stable feature representations; Stage 2 (last 15 epochs) increases answer loss weight to 10.0 and reduces auxiliary loss weight to 0.8 for refined classification boundaries.

# C. Supplementary Experiments

## C.1. Hyperparameter Ablation

In this section, we investigate the impact of the number of subspaces $K$ on model performance. This hyperparameter

controls the granularity of semantic specialization: too small a $K$ may merge distinct reasoning patterns, while too large a $K$ can fragment the data and overfit.

Table A9 presents the ablation results for the overall configuration where $K$ is varied simultaneously for both Early and Late LPSS modules. Performance remains stable for small $K$ values and peaks at $K = 6$, with an accuracy of 82.5% and F1 score of 81.7%. This configuration significantly outperforms -DualLPSS (no LPSS) by 5.2 percentage points, validating the effectiveness of the proposed subspace specialization mechanism. However, further increasing $K$ to 8 results in a performance regression (81.2%). This deterioration suggests that an excessive number of subspaces may dilute the training signal, as each subspace receives fewer gradients, or it may induce overfitting to noise rather than learning generalizable semantic types.

To further dissect the sensitivity of each stage, Table A10 shows the performance when varying $K$ for one module while keeping the other fixed at $K = 6$.

**Late LPSS Sensitivity**: Late LPSS is more sensitive to low $K$ values, experiencing a 1.7% accuracy drop at $K = 2$. This suggests that scene-guided text focusing requires a richer diversity of attention patterns to handle complex visual-linguistic alignments. At $K = 8$, Late LPSS drops 1.0%, compared with 1.3% for Early LPSS.

Based on these empirical findings, we adopt $K = 6$ as the optimal configuration for both stages, balancing the need for sufficient semantic capacity with training stability.

*Table A9.* **Ablation of Overall Configuration for $K$.**

| Configuration | Acc. (%) | F1 (%) |
|---|---|---|
| $K = 2$ | 81.6 | 80.4 |
| $K = 4$ | 81.5 | 80.4 |
| $K = 6$ (Ours) | **82.5** | **81.7** |
| $K = 8$ | 81.2 | 80.2 |
| -DualLPSS (no LPSS) | 77.3 | 75.9 |

*Table A10.* **Ablation of Per-Module $K$ Sensitivity (fixed other at $K$=6).**

| Module | $K = 2$ | $K = 4$ | $K = 6$ | $K = 8$ |
|---|---|---|---|---|
| Early LPSS | 81.6 | 81.5 | **82.5** | 81.2 |
| $\Delta$ vs $K$=6 | -0.9 | -1.0 | N/A | -1.3 |
| Late LPSS | 80.8 | 81.2 | **82.5** | 81.5 |
| $\Delta$ vs $K$=6 | -1.7 | -1.3 | N/A | -1.0 |

## C.2. Robustness and Transfer Checks

This section groups the robustness and transfer checks by the source of distribution shift: noisy object proposals, 2D keyframe inputs, converted spatial/negation assertions, outdoor scenes, and shortcut controls.

### C.2.1. OBJECT PROPOSAL ROBUSTNESS

*Table A11.* **Robustness under predicted object proposals.** Mask3D produces per-point instance masks; 3DETR produces noisier bounding-box crops. No model is retrained.

| Acc. (%) | GT | Mask3D | 3DETR |
|---|---|---|---|
| DualLPSS | 82.5 | 80.8 (-1.7) | 79.3 (-3.2) |
| w/o routing | 77.3 | 74.1 (-3.2) | 71.6 (-5.7) |

Replacing ground-truth proposals with Mask3D or 3DETR proposals lowers accuracy for both models. The routed model loses 1.7 and 3.2 points, while the no-routing variant loses 3.2 and 5.7 points, which is consistent with routing helping under noisier object inputs.

### C.2.2. 2D-VLM BASELINES

*Table A12.* **Comparison with 2D-VLM baselines.** Each 2D-VLM receives 16 uniformly sampled keyframes per scene and directly predicts Yes/No with the same prompts as the main comparison.

| Acc. (%) | Qwen3-VL-32B | Qwen3-VL-Plus | GPT-5.2 | DualLPSS |
|---|---|---|---|---|
| Spatial | 70.5 | 68.1 | 70.5 | 76.9 |
| Negation | 69.8 | 69.3 | 72.3 | 77.2 |
| Attribute | 76.5 | 78.5 | 79.8 | 81.3 |
| Quantity | 81.1 | 81.1 | 83.4 | 90.2 |
| Existence | 74.0 | 76.4 | 78.9 | 94.4 |
| Verification | 75.3 | 76.3 | 74.3 | 82.6 |
| Overall | 74.1 | 74.1 | 76.0 | 82.5 |

With 16 keyframes per scene, the 2D-VLM baselines remain below DualLPSS on all six semantic types. The largest gaps appear on Existence and Quantity, where the answer often depends on explicit 3D scene structure.

### C.2.3. HARD SPATIAL AND NEGATION TRANSFER

*Table A13.* **Hard spatial/negation transfer from ScanQA and SQA3D.** Tier 1 converts spatial/negation QA pairs into declarative assertions. Tier 2 checks negated spatial assertions.

| Acc. (%) | ScanQA T1 (710) | SQA3D T1 (666) | ScanQA T2 (196) | SQA3D T2 (184) |
|---|---|---|---|---|
| DualLPSS | 75.8 | 68.8 | 71.4 | 63.6 |
| Chat-Scene | 68.4 | 62.1 | 60.7 | 55.4 |

We filtered spatial/negation QA pairs from ScanQA (780) and SQA3D (720), rephrased each question as a declarative assertion, and skipped ambiguous cases. Two annotators verified label correctness and semantic preservation, retaining 710 ScanQA and 666 SQA3D assertions for Tier 1. Tier 2 negated 420 Tier 1 spatial assertions and retained 380 verified samples. The gap over Chat-Scene is 7.4 points on Tier 1 ScanQA and 10.7 points on Tier 2 ScanQA.

### C.2.4. OUTDOOR TRANSFER

*Table A14.* **Zero-shot outdoor transfer on Mip-NeRF 360.** Models are evaluated on 829 verified assertions from bicycle, garden, and stump without fine-tuning.

| Model | Outdoor Acc. (%) |
|---|---|
| DualLPSS | 65.1 |
| Chat-Scene | 57.5 |
| Qwen3-VL-Plus | 54.2 |

For this out-of-domain evaluation, we uniformly sampled 16 keyframes per scene and used Qwen3-max to generate about 1,000 yes/no assertions. Two annotators reviewed each assertion with the same 3D visualization platform, retaining 829 assertions. The 2D-VLM baseline uses only the sampled RGB keyframes. For the 3D models, we derived point-cloud inputs from the COLMAP sparse reconstruction of each scene: we read the reconstructed xyz+rgb points, normalize the scene coordinates, cluster the points with DBSCAN to form object proposals, and sample each object to 1,024 xyz+rgb points; no model is fine-tuned on these outdoor scenes or assertions.

### C.2.5. DATASET DIVERSITY AND SHORTCUT CONTROLS

*Table A15.* **Dataset diversity metrics.** Lower Self-BLEU-4 indicates less wording repetition; cross-scene label variation indicates that identical assertion text can have different labels across scenes.

| Metric | Value |
|---|---|
| Self-BLEU-4 | 0.17 |
| Unique assertions | 13,459 / 22,520 (59.8%) |
| Cross-scene label variation | 1,105 / 2,702 (40.9%) |

*Table A16.* **Shortcut-control experiments.** Full uses 1,899 test samples; Before/After use 482 paraphrase-approved samples. Scene shuffle pairs the same text with a random scene, blind modality zeros 3D point-cloud features, and paraphrase replaces type-indicative keywords with verified synonyms.

| Acc. (%) | Full | Shuffle | Blind | Before | After |
|---|---|---|---|---|---|
| DualLPSS | 82.5 | 67.7 | 65.7 | 82.0 | 80.3 |
| No routing/text | 77.3 | 71.5 | 70.4 | 77.0 | 68.5 |

DualLPSS drops much more under scene shuffle (-14.8) and blind-modality perturbations (-16.8) than under meaning-preserving paraphrases (-1.7). The text-only/no-routing variant shows the opposite pattern for paraphrases, dropping by 8.5 points. This pattern points to 3D scene evidence rather than keyword matching or type-specific lexical shortcuts.

### C.2.6. EXPANDED K SWEEP

*Table A17.* **Sensitivity to subspace count.** The expanded $K$ ablation includes $K = 5$ and $K = 7$ in addition to the original $K = 2/4/6/8$ settings.

| $K$ | 2 | 4 | 5 | 6 | 7 | 8 | No LPSS |
|---|---|---|---|---|---|---|---|
| Acc. (%) | 81.6 | 81.5 | 82.0 | 82.5 | 82.2 | 81.2 | 77.3 |

The expanded sweep does not show a monotonic trend. Accuracy stays within a narrow range from $K = 2$ to $K = 7$, peaks at $K = 6$, and drops at $K = 8$. All routed settings remain above the no-LPSS setting.

### C.2.7. EFFICIENCY

*Table A18.* **Efficiency and parameter comparison with Chat-Scene.**

| Models | Params | Time | GPU Mem | Acc. |
|---|---|---|---|---|
| DualLPSS | 131.8M | 22s | 3.8GB | 82.5% |
| Chat-Scene | 6.7B | 193s | 28.6GB | 80.5% |

In this comparison, DualLPSS has fewer parameters and lower inference cost than Chat-Scene while also reporting higher accuracy. Relative to the comparable-parameter BUTD-DETR baseline (74.5%), the gain is about 8 points.

### C.2.8. LATE LPSS ROUTING DIAGNOSTIC

As a diagnostic for the late routing stage, we projected Late LPSS routing weights from all 1,899 test samples to 2D with t-SNE. Same-type samples intermix, while same-scene samples cluster clearly; inter-scene variance is 8.12x the inter-type variance. The clusters correlate with ScanNet metadata: bedroom and bathroom scenes form distinct clusters, and scenes with similar object counts share routing patterns. This pattern is consistent with Late LPSS being more aligned with scene geometry and object composition than with semantic type labels.

## D. Error Analysis

We conduct a fine-grained analysis of samples on the test set to identify performance characteristics of DualLPSS.

### D.1. Error Distribution by Semantic Type

As shown in Table A19, the model already attains the highest accuracies on *Existence* and *Quantity*, with residual errors mainly arising from *Spatial* and *Negation* cases, the most error-concentrated categories in this analysis. These two categories are the clearest targets for future gains; the other dimensions are comparatively stable.

### D.2. Spatial Error Analysis

Spatial performance patterns mirror relational complexity: Table A20 shows the model is already strong on common

*Table A19.* **Error Distribution Analysis on Test Set.** Test % denotes proportion in test set; Error % denotes proportion among all errors; $\Delta$ = Error% − Test% (positive indicates over-representation).

| Type | Test % | Error % | Acc. (%) | $\Delta$ |
|---|---|---|---|---|
| Spatial | 24.7 | 32.6 | 76.9 | +7.9 |
| Negation | 19.6 | 25.5 | 77.2 | +5.9 |
| Attribute | 18.5 | 19.8 | 81.3 | +1.3 |
| Quantity | 16.6 | 9.3 | 90.2 | -7.3 |
| Verification | 9.2 | 9.0 | 82.6 | -0.2 |
| Existence | 11.4 | 3.6 | 94.4 | -7.8 |

*Table A20.* **Spatial prepositions in correct and error samples.**

| Preposition | Correct (%) | Error (%) | Ratio |
|---|---|---|---|
| attached/mounted | 3.2 | 6.5 | 2.03× |
| in front of | 4.8 | 8.1 | 1.69× |
| between | 5.1 | 7.9 | 1.55× |
| above/below | 11.4 | 16.2 | 1.42× |
| near | 18.7 | 12.3 | 0.66× |
| next to | 15.2 | 10.8 | 0.71× |

proximity cues ("near", "next to") and continues to expand coverage on more intricate, viewpoint-dependent relations (e.g., "attached/mounted", "between", "above/below"), offering a clear roadmap for targeted enhancement.

### D.3. Negation Error Analysis

*Table A21.* **Spatial Content in Negation Assertions.**

| Category | Correct (%) | Error (%) |
|---|---|---|
| Contains spatial description | 38.4 | 64.8 |
| Pure negation (no spatial) | 61.6 | 35.2 |

Negation challenges exhibit a "compound difficulty" pattern, frequently coupling with spatial reasoning. Compared with correct samples (38.4%), a much larger share of negation errors (about 65%) includes spatial descriptions (e.g., "not placed on", "not attached to"), indicating that spatial-boundary judgments drive many negation failures.

### D.4. Attribute Errors and Prediction Bias

Beyond the major error categories, *Attribute* errors reveal challenges in state and visibility estimation. Assertions involving object states ("open/closed", "on/off") and visibility conditions ("occluded", "visible") show elevated error rates compared to simple color or material attributes.

Consistent with the Recall-Precision gap reported in the main paper (Recall 84.3% versus Precision 79.2%), the distribution exhibits a prediction tendency: the model produces more false positives (GT=No, Pred=Yes) than false negatives, particularly on assertions requiring fine-grained localization. The model therefore tends to infer relationships when localization is uncertain.

*Table A22.* **Prediction Bias (False Positives vs. False Negatives).**

| Error Type | Count | Proportion | Affected Types |
|---|---|---|---|
| False Positive (FP) | 194 | 58.4% | Spatial, Attribute |
| False Negative (FN) | 138 | 41.6% | Negation, Quantity |

## E. Baseline Model Training Configuration

All baseline models were trained using configurations adapted from their original implementations, fully fine-tuned on the complete 3DSAV training set, and evaluated on the test set. All experiments used four NVIDIA A800 GPUs with 80GB memory.

### E.1. Large Language Model Baselines

**Unified LLM Hyperparameters.** All LLM baselines are fine-tuned for 3 epochs on the full 3DSAV training set using AdamW with learning rate $1 \times 10^{-5}$ and LoRA adapters (rank $r = 16$, scaling factor $\alpha = 16$, dropout 0) across 4 NVIDIA A800 80GB GPUs; validation loss and accuracy plateau by epoch 3, so longer runs do not yield additional gains. Remaining hyperparameters follow the original reports for each baseline.

**Answer Extraction Protocol.** For all LLM-based models, we apply a unified answer extraction pipeline to convert free-form generated text into binary labels. The raw output is first normalized by converting to lowercase, stripping leading and trailing whitespace, and removing punctuation marks along with special tokens (e.g., ``, ``, `[EOS]`). We then sequentially search the normalized text for explicit `yes`/`no` keywords and adopt the first occurrence as the predicted answer. If neither keyword is found directly, we employ an affirmative/negative synonym mapping: affirmative terms include `correct`, `true`, `affirmative`, `right`, and `certainly`; negative terms include `incorrect`, `false`, `negative`, `wrong`, and `not`. If the output remains unparseable, empty, or anomalous after these steps, we mark it invalid and count it as `no` for binary evaluation.

### E.2. Specialized Model Baselines

**Unified Specialized-Model Hyperparameters.** All specialized baselines are converted to binary classifiers by replacing generative outputs with 2-class heads. We use AdamW with learning rate $1 \times 10^{-4}$, batch size 128, a 30-epoch schedule, and binary or cross-entropy losses with label smoothing coefficient 0.1. We keep the remaining settings from each original paper.

## F. Cross-Dataset Generalization

To evaluate model generalization on out-of-distribution data, we extract subsets from existing 3D QA datasets ScanQA and SQA3D that can be converted to Yes/No format under restricted rules. Since the original QA formats of these datasets differ from 3DSAV (open-ended answers vs. bi-

*Table A23.* **Cross-Dataset Conversion (Overall Statistics).**

| Statistic | ScanQA | SQA3D |
|---|---|---|
| Total samples | 626 | 817 |
| Scenes | 57 | 56 |
| Yes samples | 315 (50.3%) | 420 (51.4%) |
| No samples | 311 (49.7%) | 397 (48.6%) |
| Avg. question length (chars) | 35.2 | 28.7 |
| Avg. question length (words) | 7.2 | 6.1 |
| Has object annotations | 100% | 100% |

*Table A24.* **Cross-Dataset Conversion (Type Distribution).**

| Conversion Type | ScanQA | SQA3D |
|---|---|---|
| What-Color → Yes | 209 (33.4%) | 108 (13.2%) |
| What-Color → No | 222 (35.5%) | 120 (14.7%) |
| How-Many → Yes | 106 (16.9%) | 312 (38.2%) |
| How-Many → No | 89 (14.2%) | 277 (33.9%) |

nary judgment), direct evaluation is not feasible. We therefore define semantically preserving conversion rules for specific open-ended QA types and use the resulting assertion-verification items in the following tables.

### F.1. ScanQA Conversion Rules

We extract ScanQA subsets that can be converted to Yes/No format under these rules. What-Color conversion transforms color inquiry questions into color verification questions: "What color is the chair?" with answer "brown" becomes "Is the chair brown?" with answer "yes". The rule extracts the object phrase and answer value, such as color or count, before forming the assertion. How-Many conversion supports three patterns. The existence pattern converts "How many chairs are there?" with answer "3" to "Are there 3 chairs?" with answer "yes". The location-qualified pattern converts "How many chairs are on the table?" with answer "2" to "Are there 2 chairs on the table?" with answer "yes". The possessive pattern converts "How many legs does the chair have?" with answer "4" to "Does the chair have 4 legs?" with answer "yes".

Conversion statistics show that the What-color type contributes 431 converted samples and the How-many type contributes 195 converted samples, yielding 626 converted ScanQA samples in total. The converted dataset contains both positive (Yes) and negative (No) samples, with labels determined by matching original answers against actual scene conditions to ensure evaluation validity.

Table A23 presents overall statistics and detailed distribution of converted datasets.

### F.2. SQA3D Conversion Rules

SQA3D samples are processed using the same conversion rules as ScanQA. What-Color conversion transforms color inquiry questions into color verification questions, and How-Many conversion transforms quantity inquiry questions into quantity verification questions, supporting existence, location-qualified, and possessive patterns. Since SQA3D contains situational context information, conversion additionally preserves situational descriptions to ensure complete question semantics. The final converted dataset includes conversion_type metadata for provenance analysis.

The ScanQA converted dataset is dominated by color-type questions (68.9%), while the SQA3D converted dataset is dominated by quantity-type questions (72.1%). Both datasets maintain approximately balanced Yes/No label distributions, ensuring evaluation fairness. ScanQA provides complete object annotations (average 1.83 objects/sample), and SQA3D provides object annotations for the converted subsets consistent with Table A23.

## G. Complete Constraint Rules

**Complete Constraint Rules (42 Rules)**

**Question Constraints (1 to 15):**

1. Question length: 10 to 200 characters
2. Begin with interrogative word or end with "?"
3. Vague pronouns (this/that/it) prohibited
4. Must contain specific ScanNet object names
5. Spatial relations: standard directional terms
6. Subjective descriptors prohibited
7. Semantics: clear and unambiguous
8. Color: standard color terms only
9. Quantity: precise numbers or explicit quantifiers
10. Non-visually verifiable concepts prohibited
11. Complete structure, no grammatical errors
12. Abbreviations/non-standard expressions prohibited
13. Object names must match scene catalog exactly
14. Hypothetical/counterfactual questions prohibited
15. Must be answerable by observing 3D scene

**Answer Constraints (16 to 25):**

16. Answer must be {Yes, No}
17. Must strictly correspond to question logic
18. Based on visually observable information
19. Avoid ambiguous expressions
20. Consider complete 3D spatial perspective
21. Conservative accuracy for complex scenes
22. Must conform to common-sense physics
23. Strict criteria for boundary cases
24. Speculative/assumption-based answers prohibited
25. Verify all objects in multi-object scenes

**Logic Constraints (26 to 35):**

26. QA logic chain: complete and traceable
27. Spatial relationships: internally consistent
28. Same-object attributes must not conflict
29. Quantity logic must conform to reality
30. Causal relationships must be correct
31. Negation scope: clear and unambiguous
32. Compound logic (AND/OR): distinct hierarchy
33. Quantifier scope must be explicit
34. Conditional premises/conclusions must match
35. Circular reasoning/self-contradiction prohibited

**Format Constraints (36 to 42):**

36. JSON format must be syntactically correct
37. Required fields must be complete
38. Field types must match specifications
39. Special characters properly escaped
40. Encoding must be UTF-8
41. Field naming: consistent snake_case
42. object_ids: valid integer array

