# OpenReview forum: "3D Scene Assertion Verification"
_ICML.cc/2026/Conference — ICML 2026 regular_

### Official Review · Reviewer_7Q1s · 2026-03-10

**Soundness:** 2
**Presentation:** 2
**Significance:** 2
**Originality:** 2
**Overall Recommendation:** 4
**Confidence:** 3

**Summary:**

This paper introduces 3D Scene Assertion Verification, a reformulation of 3D scene understanding as binary truth-value prediction over natural-language assertions, and presents 3DSAV, a benchmark built on ScanNet with 22,520 human-annotated assertions over 682 scenes and six semantic types. To address this task, the paper proposes DualLPSS, a dual-stage routing framework that injects type-aware biases before cross-modal fusion and scene-aware biases during text aggregation. On 3DSAV, the method achieves 82.5 Acc / 81.7 Macro-F1, outperforming the strongest reported baseline Chat-Scene by about 2.0 points in accuracy.

**Compliance With Llm Reviewing Policy:**

Affirmed.

**Final Justification:**

My main concerns have been addressed satisfactorily, particularly through the additional experiments on shortcut robustness and harder transfer settings. While some limitations remain, the rebuttal significantly strengthens the submission, and I will raise my score accordingly.

**Key Questions For Authors:**

1. Can the authors better clarify in what sense this is a genuinely new task rather than a structured subset or reformulation of existing yes/no 3D-QA?
2. Can the authors provide experiments that reduce the recoverability of semantic type from lexical cues, to better test whether DualLPSS is learning real 3D logical reasoning rather than type-triggered routing?
3. Can the authors evaluate transfer on harder converted settings beyond color/counting, especially negation and compositional spatial verification?

**Limitations:**

yes

**Strengths And Weaknesses:**

### Strengths

1. The paper is well motivated. The authors argue that standard generative 3D-VQA is not ideal for downstream decision-making because free-form descriptive outputs can be ambiguous, whereas binary verification is more directly aligned with logical execution and embodied decision loops. This motivation is clear and practically meaningful.
2. A major strength is the benchmark construction. 3DSAV is reasonably large, uses human verification, and is explicitly designed to cover multiple semantic types such as spatial, negation, attribute, quantity, and existence. The benchmark also emphasizes more complex logical structures than prior 3D-QA datasets, including much higher rates of negation, disjunction, and multi-object reasoning. The supplement further reports careful annotation with at least two annotators per sample and substantial agreement statistics.
3. The method is also sensible. DualLPSS introduces two routing stages: an Early LPSS that adjusts cross-modal fusion based on question semantics, and a Late LPSS that adjusts textual aggregation based on scene content. The ablations show that both stages help, with the full model improving from 77.3 to 82.5 accuracy over the backbone, and each single stage already contributing substantial gains.

### Weaknesses
1. My main concern is that the task novelty is somewhat overstated. While the paper frames this as a new task, binary yes/no reasoning is not absent from prior 3D-QA settings. In my view, this contribution is better understood as a task reformulation and benchmark refocusing toward assertion verification, rather than a fundamentally new capability. The paper does provide a cleaner verification-oriented setup, but the conceptual gap from existing 3D-QA is smaller than the presentation suggests.
2. The strongest contribution is the benchmark, not the method. DualLPSS is reasonable, but methodologically it feels like a benchmark-aligned routing design rather than a broadly novel modeling principle. In particular, both stages use K = 6 subspaces, explicitly matching the six semantic types, and the routing visualization shows clear type-to-subspace correspondence for the early routing stage. This makes the method feel somewhat engineered around the benchmark taxonomy.
3. The empirical gain over the best baseline is not especially large on the main benchmark: 82.5 vs. 80.5 accuracy over Chat-Scene. That is a real gain, but not one that by itself strongly establishes a major methodological advance.
4. I am not fully convinced that the paper rules out dataset-specific shortcuts. The supplement states that semantic types are assigned by a priority-based rule system using explicit keywords such as “not”, “at least”, “is there”, color/material words, and spatial relation keywords. This raises the possibility that some gains come from exploiting lexical patterns and type-specific biases, rather than from deeper joint 3D reasoning. The routing analysis, while interesting, arguably reinforces this concern because the early module aligns strongly with semantic type.
5. The cross-dataset generalization evidence is not fully convincing. The reported transfer results are based on converting only color and counting questions from ScanQA and SQA3D into verification format. That is useful, but much narrower than the core claimed strengths of the benchmark, especially negation and more compositional logical verification.

---

> ### Author Rebuttal · Authors · 2026-03-30
>
> We thank Reviewer 7Q1s for recognizing the strong benchmark construction and well-motivated formulation.
>
> **W1/Q1 (Task novelty).** We believe the reviewer may be equating the occasional yes/no answers in prior 3D-QA with 3DSAV's purpose-built verification task, but the two tasks fundamentally evaluate different capabilities.
>
> First, the two tasks solve different problems: 3D-QA is open-ended generation; 3DSAV is verification where the model judges a given statement true or false.
>
> Second, the two tasks evaluate different capabilities. Empirical results show that QA and verification performance do not positively correlate. Table 2 shows that Chat-Scene, the overall strongest at 80.5%, achieves only 69.8% on negation, below Video3D-LLM at 73.6% negation despite its lower 70.3% overall. This rank reversal suggests that QA and verification abilities are not equivalent.
>
> Third, many established tasks originated from reframing existing tasks to evaluate different capabilities. The shift from QA to Natural Language Inference (NLI) is a well-known precedent: SNLI (Bowman et al., 2015) and MultiNLI (Williams et al., 2018) established premise-hypothesis logical judgment as an independent task, central to GLUE and instrumental to BERT pre-training. NLI and QA are superficially similar yet evaluate different reasoning abilities; 3DSAV introduces a similar paradigm to 3D scene understanding for the first time, analogous to the QA-to-NLI transition.
>
> Fourth, our task has strong practical value: downstream applications such as AR placement, robotic grasping, and navigation require binary decisions. QA output needs additional parsing for decision-making, whereas verification output directly drives control.
>
> Furthermore, our contributions extend beyond task definition to also include the 3DSAV benchmark and the DualLPSS method.
>
> **W2 (K=6 and routing design).** The model does not receive type labels during training or inference; routing weights are computed from question text and scene representations. The correspondence in Figure 4 formed naturally during training, with no preassigned mapping. We added K=5 and K=7 to the existing ablation in Appendix C:
>
> |K|2|4|5|6|7|8|none|
> |:-|:-:|:-:|:-:|:-:|:-:|:-:|:-:|
> |Acc(%)|81.6|81.5|82.0|82.5|82.2|81.2|77.3|
>
> Performance is flat (81.2%-82.5%) with K=6 best, showing K-insensitivity and stable routing gains.
>
> **W3 (Empirical gain).** The DualLPSS vs. Chat-Scene comparison is not on equal footing:
>
> |Models|Params|Time|GPU Mem|Acc|
> |:-|:-:|:-:|:-:|:-:|
> |DualLPSS|131.8M|22s|3.8GB|82.5%|
> |Chat-Scene|6.7B|193s|28.6GB|80.5%|
>
> DualLPSS outperforms Chat-Scene with far fewer resources. Our method also achieves ~8 pp gain over comparable-parameter BUTD-DETR (74.5%), demonstrating our method's gain is not marginal.
>
> **W4/Q2 (Shortcut concerns).** We conducted two experiments to test whether the model relies on genuine 3D reasoning. Exp A: with architecture and weights unchanged, we ran two perturbations on all 1,899 test samples: scene shuffle keeps text unchanged but replaces the paired 3D scene with a random one; blind modality zeros all 3D point cloud features while keeping text. Exp B: we sampled 100 assertions from each of the six types, totaling 600, and replaced type-indicative keywords with synonyms, e.g., "near" to "in the vicinity of." Each paraphrase was reviewed by two annotators and retained only if both agreed meaning was preserved, yielding 482 assertions.
>
> |Acc(%)|Full(1899)|Shuffle|Blind|Before(482)|After|
> |:-|:-:|:-:|:-:|:-:|:-:|
> |DualLPSS|82.5|67.7(-14.8)|65.7(-16.8)|82.0|80.3(-1.7)|
> |w/o routing / Text-only|77.3|71.5(-5.8)|70.4(-6.9)|77.0|68.5(-8.5)|
>
> DualLPSS drops far more under 3D perturbation yet far less under keyword replacement, indicating genuine 3D utilization rather than lexical shortcuts. Type labels are never used as model training input; they serve only for per-type evaluation (Appendix A.2).
>
> **W5/Q3 (Cross-dataset transfer).** We extended to harder spatial and negation types. We filtered spatial/negation QA pairs from ScanQA (780) and SQA3D (720), rephrased each question into a declarative assertion; labels were determined by original answer semantics (affirmative→yes, negative→no), with ambiguous cases skipped. Two annotators independently verified each assertion, checking label correctness and semantic preservation; only assertions approved by both were retained, yielding 710 from ScanQA and 666 from SQA3D (Tier 1). We further constructed Tier 2 by negating 420 Tier 1 spatial assertions, e.g., "the chair is near the table" to "the chair is NOT near the table." Two annotators verified grammar and semantic opposition, yielding 380 samples. Results with paper checkpoints:
>
> |Acc(%)|ScanQA T1(710)|SQA3D T1(666)|ScanQA T2(196)|SQA3D T2(184)|
> |:-|:-:|:-:|:-:|:-:|
> |DualLPSS|75.8|68.8|71.4|63.6|
> |Chat-Scene|68.4|62.1|60.7|55.4|
>
> DualLPSS's advantage grows from 7.4 pp (Tier 1) to 10.7 (Tier 2), further ruling out dataset shortcuts. New results in Appendix.

---

> > ### Author Rebuttal · Reviewer_7Q1s · 2026-04-03
> >
> > Thank you for the detailed rebuttal. My main concerns have been addressed satisfactorily, particularly through the additional experiments on shortcut robustness and harder transfer settings. While some limitations remain, the rebuttal significantly strengthens the submission, and I will raise my score accordingly.

---

> > > ### Author Response · Authors · 2026-04-05
> > >
> > > Thank you for the detailed reading of our rebuttal and for updating the score. The feedback throughout this discussion has been very helpful. The supplementary experiments from this discussion will be incorporated into the revision (Appendix C and F).

---

### Official Review · Reviewer_NKck · 2026-03-10

**Soundness:** 3
**Presentation:** 3
**Significance:** 3
**Originality:** 3
**Overall Recommendation:** 4
**Confidence:** 3

**Summary:**

This paper introduces 3D Scene Assertion Verification, a new task for 3D scene understanding in which a model must determine whether a natural-language assertion about a 3D scene is true or false. The authors motivate this task by arguing that existing 3D visual question answering systems are predominantly generative, which can yield descriptive but non-decisive outputs that are not suitable for downstream decision-making in embodied AI and robotics. In contrast, the proposed task requires deterministic binary judgments.

To support this task, the paper presents 3DSAV, a benchmark built on ScanNet with 22,520 assertions across 682 scenes, covering six semantic types: spatial, negation, attribute, quantity, existence, and verification. The dataset is constructed using a two-stage pipeline: LLM-based candidate generation followed by human verification and filtering, with each assertion reviewed by at least two annotators.

The paper also proposes DualLPSS, a method tailored to this task. DualLPSS uses a dual-stage routing mechanism: an Early LPSS module that adapts cross-modal fusion based on question semantics, and a Late LPSS module that adapts text aggregation based on scene content. Experiments show that DualLPSS outperforms a range of prior 3D-VQA and 3D-LLM baselines on 3DSAV, and ablations suggest both routing stages contribute to performance improvements.

**Compliance With Llm Reviewing Policy:**

Affirmed.

**Key Questions For Authors:**

# 1. Comparison to strong 2D / multi-view VLM baselines
Since 3DSAV is built on ScanNet, could the authors evaluate strong 2D-VLM or video-VLM baselines on a subset of the benchmark using RGB images or multiview renderings? This would help clarify which parts of the benchmark genuinely require 3D reasoning and which can already be handled by 2D models. A strong result here could either strengthen the paper’s claims or reveal useful limitations of current 2D-VLMs.

# 2. Dataset generation bias and linguistic diversity
Because candidate assertions are first generated by an LLM and then filtered by humans, how diverse are the final assertions linguistically? Are there repeated templates or systematic lexical patterns that models may exploit? Some statistics or qualitative analysis on wording diversity would help.

# 3. Generalization beyond ScanNet
Do the authors expect DualLPSS and 3DSAV-style supervision to transfer to other 3D scene datasets or embodied 3D benchmarks? Some discussion of transferability beyond indoor ScanNet scenes would strengthen the significance claim.

**Limitations:**

Partially.
The paper does include a short limitations/future work paragraph discussing remaining failure modes and the possible need for interactive view acquisition, which is a useful start.
Meanwhile, the discussion could be improved by being more explicit about:
- The benchmark’s restriction to indoor ScanNet scenes
- Possible generation bias introduced by LLM-assisted assertion construction
- The absence of comparison to strong 2D-VLM baselines,
- The fact that verification is limited to binary truth judgments, which may not cover uncertainty estimation or ambiguity present in real-world downstream systems.

**Strengths And Weaknesses:**

# Strengths
## 1. Clear and novel task formulation
The task itself is a meaningful contribution. Moving from free-form 3D VQA to binary assertion verification is well motivated, especially for downstream uses that require explicit, reliable decisions rather than descriptive outputs. This framing is intuitive and potentially useful for robotics, embodied agents, and safety-critical decision loops.

## 2. Useful benchmark design
The proposed 3DSAV benchmark is reasonably sized and appears thoughtfully constructed. The six semantic types provide a useful diagnostic breakdown, especially because the dataset emphasizes logical phenomena such as negation, disjunction, and quantity more strongly than prior 3D QA datasets. The near-balanced label distribution is also a positive design choice.

## 3. Method is reasonably aligned with the task
The proposed DualLPSS architecture is task-aware rather than arbitrary. The idea of using early routing for question-type-dependent fusion and later routing for scene-dependent focus is sensible for assertion verification, and the ablation results support that both modules help.

# Weaknesses
## 1. Limited comparison to 2D-VLMs / multi-view VLMs
A notable missing baseline class is strong 2D or video VLMs operating on ScanNet RGB frames or multiview renderings. Since ScanNet contains RGB data, comparison against strong 2D-VLMs could help clarify which aspects of the task truly require 3D reasoning versus what can already be solved from image/video evidence. Even if 2D models are not suitable for all samples, partial comparison would strengthen the paper considerably.
## 2. The cross-dataset generalization setup is somewhat limited
The reported generalization to ScanQA and SQA3D is interesting, but it relies on converting open-ended questions into verification format for only selected subsets such as color and counting. This is a reasonable first step, though it is not yet strong evidence that the model generalizes broadly beyond 3DSAV.
## 3. Some claims about generative models are somewhat overstated
The paper argues that generative VQA paradigms are fundamentally misaligned with verification. The motivation is reasonable, but some of the wording is stronger than what the experiments strictly establish. In practice, generative models can often be adapted into discriminative settings, so the paper could present this contrast in a slightly more measured way.

---

> ### Author Rebuttal · Authors · 2026-03-30
>
> We thank Reviewer NKck for the constructive feedback and for recognizing the clear task formulation and useful benchmark design.
>
> **W1/Q1 (2D-VLM baselines).** We appreciate this suggestion and conducted 2D-VLM comparison experiments. We uniformly sampled 16 keyframes per scene as image input and used the same prompts as in the main comparison experiments to have each model directly output Yes or No for each assertion. We evaluated three models: one open-source model Qwen3-VL-32B-Instruct, and two proprietary models Qwen3-VL-Plus and GPT-5.2:
>
> |Acc(%)|Qwen3-VL-32B|Qwen3-VL-Plus|GPT-5.2|DualLPSS|
> |:-|:-:|:-:|:-:|:-:|
> |Spatial|70.5|68.1|70.5|76.9|
> |Negation|69.8|69.3|72.3|77.2|
> |Attribute|76.5|78.5|79.8|81.3|
> |Quantity|81.1|81.1|83.4|90.2|
> |Existence|74.0|76.4|78.9|94.4|
> |Verification|75.3|76.3|74.3|82.6|
> |Overall|74.6|74.6|76.4|82.5|
>
> DualLPSS outperforms all three 2D-VLMs across all six types. The gap is especially large on existence (+15.5 pp) and quantity (+6.8 pp), where 3D scene understanding matters most.
>
> **W2/Q3 (Cross-dataset generalization).** Table 3 already reports color and counting cross-dataset results. We used keyword matching to filter spatial/negation QA pairs from ScanQA (780) and SQA3D (720), rephrased each question into a declarative assertion; labels were determined by the original answer semantics (affirmative→yes, negative→no), with ambiguous cases skipped. Two annotators independently verified each assertion, checking label correctness and semantic preservation; only assertions approved by both were retained, yielding 710 from ScanQA and 666 from SQA3D. Results with paper checkpoints:
>
> |Acc(%)|ScanQA(710)|SQA3D(666)|
> |:-|:-:|:-:|
> |DualLPSS|75.8|68.8|
> |Chat-Scene|68.4|62.1|
> |w/o routing|63.7|57.2|
>
> DualLPSS leads Chat-Scene by 7.4 pp on ScanQA and 6.7 on SQA3D; w/o routing falls a further 4.7-4.9 pp below Chat-Scene.
>
> We also evaluated on 3 Mip-NeRF 360 outdoor scenes (bicycle, garden, stump), none in ScanNet. We uniformly sampled 16 keyframes per scene and fed them to Qwen3-max to generate approximately 1,000 yes/no assertions. Each assertion was independently reviewed by two annotators using our 3D visualization web annotation platform; only assertions where both annotators agreed were retained, yielding 829 verified assertions. For model evaluation, we applied scene normalization to the COLMAP sparse point cloud of each scene, used DBSCAN clustering to segment object-level proposals, sampled each object to 1,024 points in 6-dim xyz+rgb, and ran DualLPSS with the paper checkpoint in a zero-shot manner:
>
> |Acc(%)|Outdoor(829)|
> |:-|:-:|
> |DualLPSS|65.1|
> |Chat-Scene|57.5|
> |Qwen3-VL-Plus|54.2|
>
> DualLPSS outperforms Chat-Scene by 7.6 pp and Qwen3-VL-Plus by 10.9 pp without fine-tuning, showing transferability to out-of-domain scenes.
>
> **W3 (Generative model claims).** We appreciate this feedback. The reviewer notes that some claims about generative models are overstated. We will revise the wording throughout the paper, e.g., replacing "generative models are unsuitable for verification" with "suboptimal in practice for structured logical verification." The empirical basis: Chat-Scene reaches a 30.2% error rate on negation (Table 2), with the typical failure being descriptive text output rather than a direct Yes/No judgment, reflecting a mismatch between generative training objectives and verification requirements.
>
> **Q2 (Linguistic diversity and bias).** We computed three diversity metrics on all 22,520 assertions: lower Self-BLEU-4 means less wording repetition; higher unique ratio means lower templatization; higher cross-scene label variation means the same assertion can be yes in one scene and no in another, so the answer depends on the scene rather than the text.
>
> |Metric|Value|
> |:-|:-:|
> |Self-BLEU-4|0.17|
> |Unique assertions|13,459/22,520 (59.8%)|
> |Cross-scene label variation|1,105/2,702 (40.9%)|
>
> Beyond dataset-level diversity, we conducted two experiments. Exp A: with architecture and weights unchanged, we ran two perturbations on all 1,899 test samples: scene shuffle keeps text unchanged but replaces the paired 3D scene with a random one; blind modality zeros all 3D point cloud features while keeping text. Exp B: we sampled 100 assertions from each of the six types, totaling 600, and replaced type-indicative keywords with synonyms, e.g., "near" to "in the vicinity of." Each paraphrase was reviewed by two annotators and retained only if both agreed meaning was preserved, yielding 482 assertions.
>
> |Acc(%)|Full(1899)|Shuffle|Blind|Before(482)|After|
> |:-|:-:|:-:|:-:|:-:|:-:|
> |DualLPSS|82.5|67.7(-14.8)|65.7(-16.8)|82.0|80.3(-1.7)|
> |w/o routing / Text-only|77.3|71.5(-5.8)|70.4(-6.9)|77.0|68.5(-8.5)|
>
> DualLPSS drops far more under 3D perturbation yet far less under keyword replacement, indicating routing drives genuine 3D utilization. All new results will be added to Appendix.

---

### Official Review · Reviewer_7BuF · 2026-03-13

**Soundness:** 2
**Presentation:** 3
**Significance:** 2
**Originality:** 2
**Overall Recommendation:** 4
**Confidence:** 2

**Summary:**

The paper introduces a novel task called 3D Scene Assertion Verification, which shifts the paradigm of 3D scene understanding from generative descriptions (like 3D-VQA) to deterministic logical verification (binary Yes/No judgments). A core challenge discussed by the manuscript is the decision ambiguity introduced by generative models, which often output declarative facts that lack the structured, actionable judgments required for downstream embodied AI and robotics planning. To address this, the authors construct 3DSAV, the first large-scale diagnostic benchmark comprising 22,520 human-verified assertions across 682 ScanNet scenes, categorized into six semantic types. Furthermore, they propose DualLPSS (Dual-stage Layered Progressive Subspace Specialization), a framework utilizing a dual-stage routing mechanism for type-aware cross-modal fusion and scene-guided assertion focusing. Overall, the authors analyze the concept of specialized reasoning mechanisms and demonstrate that DualLPSS achieves state-of-the-art performance (82.5% accuracy) while maintaining the inference efficiency of specialized models

**Compliance With Llm Reviewing Policy:**

Affirmed.

**Key Questions For Authors:**

1. How does DualLPSS perform when using raw, unsegmented scene point clouds or noisy object proposals generated by an off-the-shelf 3D detector, rather than ground-truth object bounding boxes?
2. Given the strict requirement for human verification (by 8 experts) to filter out hallucinations and ambiguities, how scalable is this pipeline for expanding the benchmark to new domains (e.g., outdoor scenes or dynamic environments)?
3. While the radar charts show clear type-to-subspace correspondence for Early LPSS, Late LPSS is described as "scene-dependent." Could you provide more qualitative insights or visualizations into exactly what visual features or scene contexts trigger specific Late LPSS subspaces?

**Limitations:**

Yes.

**Strengths And Weaknesses:**

## Strengths
1. Shifting from generative VQA to deterministic assertion verification is highly relevant for real-world embodied AI, where binary preconditions are necessary for safe and reliable robotic interactions.
2. The dataset is rigorously constructed using an LLM-generation and human-verification pipeline. The categorization into six semantic types (Spatial, Negation, Attribute, Quantity, Existence, Verification) allows for excellent fine-grained diagnostic evaluation of model capabilities.
3. The DualLPSS framework is well-motivated. The use of Early LPSS (question-driven routing) and Late LPSS (scene-driven routing) effectively handles complex logical structures like negation and spatial relations without the massive computational overhead of LLMs.
4. The paper provides extensive quantitative and qualitative comparisons against both specialized models and LLMs, supported by detailed ablation studies, cross-dataset generalization tests, and error analysis.

## Weaknesses
1. The methodology relies on PointNet++ to process object point clouds, which implies a reliance on relatively clean or ground-truth object segmentations/bounding boxes. Real-world robotic perception often deals with noisy, incomplete, or unsegmented point clouds.
2. As noted in the error analysis, the model still struggles with compound difficulties, particularly spatial negation (e.g., reasoning about spatial boundaries under a negation scope) and fine-grained visibility/state attributes.
3. The dataset and model are built upon ScanNet, which restricts the evaluation to indoor scenes. It is unclear how well the routing mechanism generalizes to outdoor environments or highly unstructured scenes.

---

> ### Author Rebuttal · Authors · 2026-03-30
>
> We thank Reviewer 7BuF for recognizing the relevance of deterministic verification to embodied AI and the rigor of the dataset construction.
>
> **W1/Q1 (Robustness under noisy proposals).** We replaced ground-truth segmentation with predictions from two off-the-shelf 3D detectors on the full test set of 1,899 samples across 67 scenes. Mask3D (Schult et al., ICRA 2023) is an instance segmentation model pretrained on ScanNet; it takes the full scene point cloud as input and outputs per-point instance masks, from which we extract each object's point cloud. 3DETR (Misra et al., ICCV 2021) is a 3D Detection Transformer pretrained on ScanNet; it also takes the full scene point cloud as input and outputs 3D bounding boxes, from which we crop each object's point cloud. Since bounding boxes are less precise than per-point masks, the crops include background and neighboring points, making 3DETR noisier than Mask3D. Per-object point clouds from both detectors are sampled to 1,024 points in 6-dim xyz+rgb and fed into DualLPSS with the same checkpoint, without any retraining:
>
> |Acc(%)|GT|Mask3D|3DETR|
> |:-|:-:|:-:|:-:|
> |DualLPSS|82.5|80.8(-1.7)|79.3(-3.2)|
> |w/o routing|77.3|74.1(-3.2)|71.6(-5.7)|
>
> DualLPSS drops only 1.7 and 3.2 pp from GT under Mask3D and 3DETR, while the no-routing variant drops 3.2 and 5.7 pp. The gap widens as noise increases (5.2→6.7→7.7 pp), indicating routing provides greater compensation as input quality degrades.
>
> **W2 (Compound difficulty).** Spatial and negation are the hardest types across all models, and this is a shared challenge in the field. DualLPSS still achieves the best performance on these hard types. Per-type accuracy from Table 2:
>
> |Acc(%)|DualLPSS|Chat-Scene|Video3D-LLM|3D-VisTA|
> |:-|:-:|:-:|:-:|:-:|
> |Spatial|76.9|73.9|62.7|66.1|
> |Negation|77.2|69.8|73.6|68.5|
> |Attribute|81.3|79.8|68.4|75.9|
> |Quantity|90.2|95.3|76.4|80.7|
> |Existence|94.4|93.7|80.1|91.9|
> |Verification|82.6|79.8|64.4|75.6|
>
> DualLPSS ranks first in five of six types, with the largest leads on the two hardest: spatial and negation. The "remaining failure mode" in the Limitation section is relative to DualLPSS's own other types, not relative to other models. Appendix D shows ~65% of negation errors involve spatial descriptions, indicating compound spatial-negation reasoning is a systematic challenge shared by all models.
>
> **W3 (Indoor/outdoor generalization).** The reviewer notes that the model is trained and evaluated solely on ScanNet indoor scenes. To address this, we constructed an out-of-domain evaluation on 3 Mip-NeRF 360 outdoor scenes: bicycle, garden, and stump, none of which appear in ScanNet.
>
> For dataset construction, we uniformly sampled 16 keyframes per scene and fed them to Qwen3-max to generate approximately 1,000 yes/no assertions across the 3 scenes. Each assertion was independently reviewed by two annotators using our custom 3D visualization web annotation platform; an assertion was retained only if both annotators judged it correct, yielding 829 verified assertions. For model evaluation, we applied scene normalization to the COLMAP sparse point cloud of each scene, then used DBSCAN clustering to segment object-level proposals, sampled each object to 1,024 points in 6-dim xyz+rgb, and ran DualLPSS with the paper's checkpoint in a zero-shot manner:
>
> |Acc(%)|Outdoor(829)|
> |:-|:-:|
> |DualLPSS|65.1|
> |Chat-Scene|57.5|
> |Qwen3-VL-Plus|54.2|
>
> DualLPSS outperforms Chat-Scene by 7.6 pp and Qwen3-VL-Plus by 10.9 pp without fine-tuning, showing that the learned 3D verification capability transfers beyond the training domain without adaptation.
>
> **Q2 (Pipeline scalability).** Our annotation was completed by 8 annotators with backgrounds in 3D computer vision, producing a total of 22,520 assertions, each independently reviewed by at least 2 annotators, as detailed in Appendix A.3. The core tool is our custom 3D visualization web annotation platform (Appendix A.3, Figure A1), supporting GT correction, sample cleaning, and grounding annotation. The outdoor evaluation in W3 already demonstrates this platform's portability: we deployed the same platform directly on the new Mip-NeRF 360 outdoor scene data, and 4 annotators completed the review of all 829 assertions without any modifications to the platform itself.
>
> **Q3 (Late LPSS visualization).** We projected routing weights from 1,899 test samples to 2D via t-SNE. Same-type samples intermix, but same-scene samples cluster clearly (inter-scene variance 8.12x inter-type). Clusters correlate with ScanNet metadata: bedroom and bathroom scenes form distinct clusters, and scenes with similar object counts share routing patterns. Late LPSS specialization is driven by scene geometry and object composition, complementing Early LPSS's type-driven routing (Figure 4). All supplementary results will be added to Appendix.

---

> > ### Author Rebuttal · Reviewer_7BuF · 2026-04-04
> >
> > Thanks for the response. All my concerns are resolved. Please include these additional discussion and experiments in you revision.

---

> > > ### Author Response · Authors · 2026-04-05
> > >
> > > Thank you for the thoughtful engagement and for confirming that the concerns have been addressed. We really appreciate the feedback throughout this discussion.
> > >
> > > We will include the additional experiments and analysis in the revision as suggested:
> > >
> > > - Mask3D / 3DETR robustness evaluation → Appendix C
> > > - Negation error breakdown and compound difficulty analysis → Appendix D
> > > - Mip-NeRF 360 outdoor zero-shot evaluation → Appendix F
> > > - Annotation platform portability → Appendix A.3
> > > - Late LPSS routing visualization (t-SNE) → Appendix C
> > >
> > > We are glad the discussion has been productive and hope it will be taken into account in the final assessment. Please let us know if any further questions come up.

---

### Decision · Program_Chairs · 2026-04-30

**Decision:**

Accept (regular)

**Comment:**

This paper introduces 3D Scene Assertion Verification, a task that shifts 3D scene understanding from open-ended generative descriptions to deterministic binary logical verification. The reviewers all appreciated the clear task formulation, the rigorous benchmark design, and the well-motivated architecture that achieves high performance while maintaining computational efficiency. Initial concerns primarily centered around the robustness of the method to noisy point clouds (Reviewer 7BuF), the absence of strong 2D-VLM baselines (Reviewer NKck), potential dataset-specific lexical shortcuts, and the extent of out-of-domain generalization (Reviewers 7Q1s and NKck).

During the rebuttal, the authors provided additional comprehensive experiments, including robustness tests with off-the-shelf 3D detectors, evaluations against strong 2D-VLMs, and zero-shot generalization on outdoor scenes. Reviewers 7BuF and 7Q1s explicitly acknowledged that their concerns were fully resolved. While Reviewer NKck did not participate in the post-rebuttal discussion, the AC finds that the authors provided a thorough and convincing response to their initial queries. Given the unanimously positive ratings and the rebuttal, the AC is happy to recommend the acceptance of this work.